# Chirality-dependent electrical transport properties of carbon nanotubes obtained by experimental measurement

Wei Su[1,2,3,6], Xiao Li[1,2,3,6], Linhai Li[1,2,3], Dehua Yang [1,3], Futian Wang[1,2,3], Xiaojun Wei[1,2,3,4], Weiya Zhou [1,2,3,4], Hiromichi Kataura [5], Sishen Xie[1,2,3,4] & Huaping Liu[1,2,3,4] ✉

Establishing the relationship between the electrical transport properties of single-wall carbon nanotubes (SWCNTs) and their structures is critical for the design of high-performance SWCNT-based electronic and optoelectronic devices. Here, we systematically investigated the effect of the chiral structures of SWCNTs on their electrical transport properties by measuring the performance of thin-film transistors constructed by eleven distinct ($n, m$) single-chirality SWCNT films. The results show that, even for SWCNTs with the same diameters but different chiral angles, the difference in the on-state current or carrier mobility could reach an order of magnitude. Further analysis indicates that the electrical transport properties of SWCNTs have strong type and family dependence. With increasing chiral angle for the same-family SWCNTs, Type I SWCNTs exhibit increasing on-state current and mobility, while Type II SWCNTs show the reverse trend. The differences in the electrical properties of the same-family SWCNTs with different chiralities can be attributed to their different electronic band structures, which determine the contact barrier between electrodes and SWCNTs, intrinsic resistance and intertube contact resistance. Our present findings provide an important physical basis for performance optimization and application expansion of SWCNT-based devices.

Single-wall carbon nanotubes (SWCNTs) have emerged as prospective candidates for next-generation electronic and optoelectronic integrated circuits (ICs) due to their extremely high carrier mobility and current-carrying capacity[1–3]. The performance of electronic devices based on either a single SWCNT or aligned array films has been reported to exceed that of silicon-based devices with the same gate length[4–9]. Theoretical predictions indicate that 3D ICs constructed from SWCNTs would be more than 1000 times faster and more energy-efficient than silicon-based ICs[10,11]. Most recently, high-performance

SWCNT computer and microprocessor were constructed from SWCNT films, further demonstrating their potential application in high-performance ICs[11–13]. Notably, the band structure of a SWCNT is determined by its atomic arrangement structure, meaning that SWCNTs have structure-tunable optical and electrical properties[1–3,14]. Therefore, establishing the relationship between the optical and electrical properties of SWCNTs and their structures is critical for the design of high-performance carbon-based electronic and optoelectronic devices.

[1]Beijing National Laboratory for Condensed Matter Physics, Institute of Physics, Chinese Academy of Sciences, Beijing 100190, China. [2]Center of Materials Science and Optoelectronics Engineering, and School of Physical Sciences, University of Chinese Academy of Sciences, Beijing 100049, China. [3]Beijing Key Laboratory for Advanced Functional Materials and Structure Research, Beijing 100190, China. [4]Songshan Lake Materials Laboratory, Dongguan, Guangdong 523808, China. [5]Nanomaterials Research Institute, National Institute of Advanced Industrial Science and Technology (AIST), Tsukuba 305-8565, Japan. [6]These authors contributed equally: Wei Su, Xiao Li. ✉e-mail: liuhuaping@iphy.ac.cn

In recent decades, intense efforts have been made to clarify the relationship between the optical and electrical properties and structures of SWCNTs[14–26]. The optical properties and chiral structure of SWCNTs have been well established. Research on the electrical properties of SWCNTs mainly focuses on their relationship with diameter[20–24]. Kauser et al. theoretically calculated the effect of chirality and diameter on the transport properties of SWCNTs by the ensemble Monte Carlo method and an iterative solution of Boltzmann's transport equation[25,26]. They demonstrate that the effect of chirality is significantly different for small diameter tubes. The electrical transport properties of SWCNTs with different diameters have been measured based on single-SWCNT field-effect transistors (FETs)[20,21]. The results show that with increasing diameter, the SWCNTs exhibit higher on-state saturation current and carrier mobility due to the formation of a lower barrier between the SWCNTs and electrodes. Recently, benefiting from the development of solution-based sorting techniques, the effect of the diameter of SWCNTs on the electrical transport properties of their thin films has been explored[22–24]. The results are consistent with those of the theoretical calculations and the experimental results based on a single SWCNT. Except for diameter, semiconducting SWCNTs can be divided into Type I and Type II according to mod $(2n + m, 3) = 1$ or 2. Meanwhile, each type of SWCNT can be divided into different families based on the value of $(2n + m)$. The SWCNTs with the same value of $(2n + m)$ belong to the same family, in which with increasing diameter, the chiral angle increases[27]. However, due to the limited types and yields of single-chirality SWCNTs that can be prepared, the dependence between the chiral structure and electrical transport properties of SWCNTs has not yet been reported.

Recently, owing to the rapid development of gel chromatography, a variety of single-chirality SWCNTs have been prepared at a large scale, which enables us to systematically explore the relationship between the electrical transport properties of SWCNTs and their chiral structures. In this work, based on films of multiple single-chirality SWCNTs separated through gel chromatography[28–30], we systematically studied the effect of the chiral structures of SWCNTs, including (6, 5), (7, 3), (7, 5), (7, 6), (8, 4), (9, 1), (9, 2), (9, 4), (10, 2), (10, 3) and (11, 1), on the electrical transport properties. The results show that the electrical transport properties of SWCNTs with diameters less than 1 nm more strongly depend on their chiral structures than on their diameters. For example, although (6, 5) and (9, 1) SWCNTs have the same diameter, (9, 1) exhibits a much higher on-state current and higher carrier mobility. Specifically, we observed obvious type- and family-dependent relationships. For Type I SWCNTs (mod $(2n + m, 3) = 1$), both the on-state current and carrier mobility increase with increasing chiral angle in the same family ($2n + m = $ constant), while for Type II species (mod $(2n + m, 3) = 2$), the results are opposite. This can be explained by the difference in their electronic band structures, which determine the contact barrier height between metal electrodes and SWCNTs, junction resistance and even intrinsic resistance. Our present results not only deepen the understanding of the electrical performance of SWCNTs but also provide important scientific guidance for the design of high-performance carbon-based electronic and optoelectronic devices.

## Results and Discussion

To study the effect of the chiral structures of SWCNTs on their electrical transport properties, eleven types of single-chirality species were used to construct thin-film transistors, which were separated by gel chromatography[28–30]. Fig. 1a, b show the optical absorption spectra and photographs of the single-chirality SWCNT solutions, from which each type of single-chirality species can be clearly identified. For each species, one main optical absorption peak in both the $S_{11}$ and $S_{22}$ regions and the unique solution color indicate that they have a high chiral purity. The chiral purities were further evaluated by fitting the

peaks of the $S_{11}$ region in the absorption spectra through PeakFit (Supplementary Fig. 1)[28]. The chiral purities of most of the species are higher than 85%. In particular, (6, 5), (7, 3), (7, 5) and (8, 4) SWCNTs exhibit a chiral purity higher than 90%. Metallic SWCNTs in each $(n, m)$ species were detected by Raman spectra with excitation wavelengths of 488, 514, and 633 nm. And the results show that no signals of metallic SWCNTs are observed (Supplementary Fig. 2). In other words, the purities of these SWCNTs are high enough that the measured electrical properties of the corresponding thin films represent their intrinsic properties. According to the above classification rules of type and family, the 11 kinds of single-chirality SWCNTs used in our study contain both types and five families of species (as shown in Fig. 1c). Therefore, the electrical transport properties of the representative SWCNT films measured in experiments can fully reflect the influence of the chiral structure of SWCNTs on their electrical properties.

The electrical transport properties of SWCNTs were measured based on their thin-film transistors (TFTs). Before the construction of TFTs, SWCNT films with controllable densities were prepared by an alkaline small molecule regulation technique[31]. As shown in Fig. 1d, based on atomic force microscopy (AFM) images of low-density SWCNT films, we statistically analyzed their length distributions (Fig. 1e). Each SWCNT has an average length of 200 to 270 nm (details are shown in supplementary Fig. 3). Obviously, there is no significant difference in the length distribution among them. It is notable that, low-density SWCNT films is not suitable for the comparison of the electrical properties of different $(n, m)$ species due to their poor electrical performances and uniformity (supplementary Fig. 4). Figure 1f and supplementary Fig. 5 shows a typical image of the high-density SWCNT films used to construct TFTs, whose linear densities were controlled at approximately $32 \pm 5$ tubes/μm (Fig. 1g), which is much higher than the percolation threshold (~10 tubes/um) (supplementary note 1, supplementary Table 1 and supplementary Fig. 6). And the films of different $(n, m)$ species show high uniformity across the substrates (supplementary Fig. 7). To reduce the effect of the surfactant adsorbed on SWCNTs on their electrical properties, sodium deoxycholate (DOC) with a concentration of 0.05% was employed to disperse various $(n, m)$ SWCNTs, which is the lowest concentration that can stably disperse SWCNTs. After film deposition, the surfactant molecules were removed by repeated cleaning with deionized water. The results show that a few DOC molecules remained on SWCNTs (supplementary Fig. 8). The SWCNT film deposition process is described in more detail in the Methods section.

As shown in Fig. 2a, top-gate SWCNT TFTs with channel length and width of 2 and 20 μm, respectively, were fabricated on SiO₂/Si substrates to measure the electrical transport properties of SWCNT films, in which 15-nm-thick HfO₂ was used as a dielectric layer. Before the deposition of the HfO₂ layer, TFTs were annealed in vacuum to remove the O₂ molecules adsorbed on the SWCNT films, which will seriously affect the intrinsic performance of SWCNT TFTs[32,33]. Fig. 2b–f show the transfer characteristics of eleven kinds of single-chirality SWCNT TFTs, which are classified by their family. For each kind of chirality, twenty TFTs were fabricated. All TFTs work well (typical output characteristics are shown in supplementary Fig. 9) and show typical bipolar characteristics, where stronger n-type features should originate from electrostatic doping of positive charges induced by oxygen vacancies in the HfO$_x$ film[34]. Importantly, the transfer curves of different devices constructed from the same-chirality SWCNT films almost overlap, indicating that the measured electrical properties are typical. By comparing the electrical properties of different-chirality SWCNT films from the same family, the on-state currents corresponding to both p-type and n-type branches can be seen to decrease with increasing chiral angle for Type II SWCNTs, while the result is the opposite for Type I SWCNTs (the dashed lines in Fig. 2 are used as guides). To exclude the effect of the adhesion strength of DOC to SWCNTs on their electrical properties, we employed achiral molecules sodium sulfate

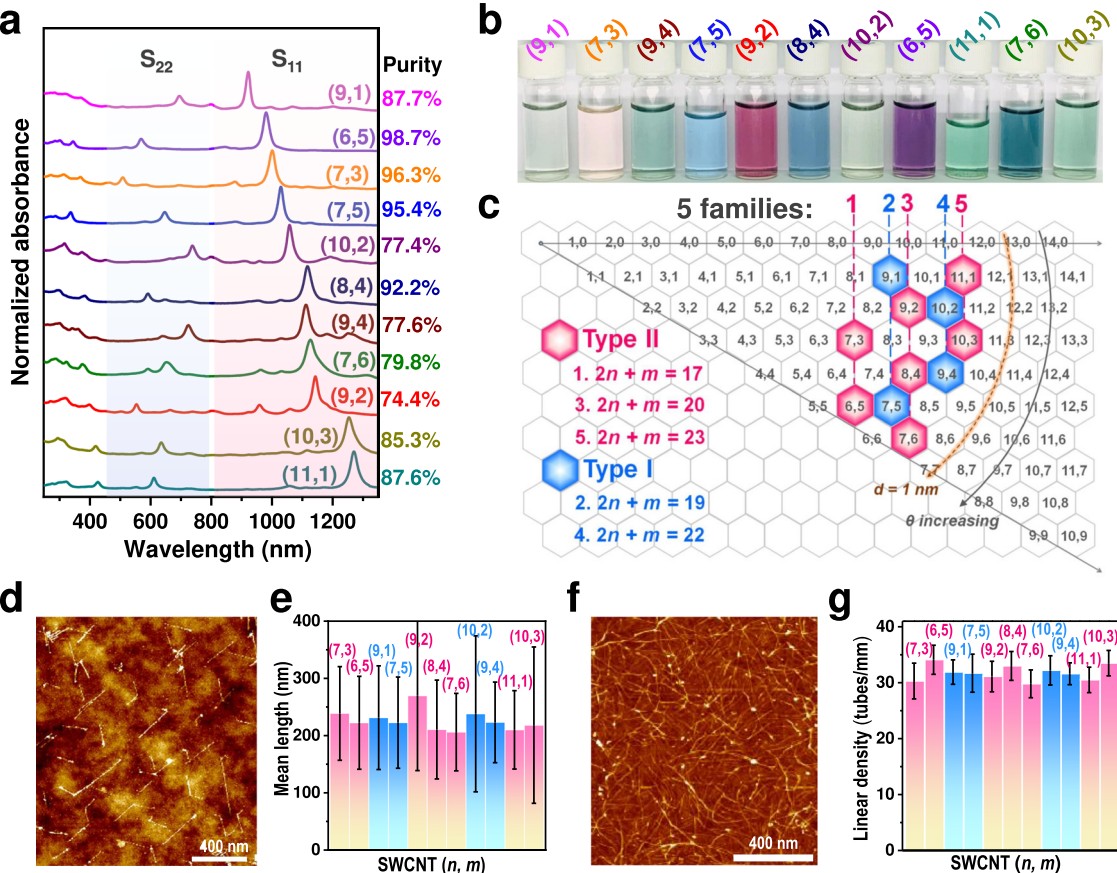

**Fig. 1 | Characterization of the single-chirality SWCNTs used in the present study. a** Optical absorption spectra of the (*n, m*) single-chirality SWCNTs. Their chiral purities are evaluated by peak fitting and indicated on the right side. **b** Photographs of the single-chirality SWCNT solutions. **c** Location of the single-chirality SWCNTs in the chiral map. **d** Typical atomic force microscopy (AFM) image of a low-density SWCNT film for length measurement. **e** Statistical distribution of the lengths of different-chirality SWCNTs. **f** Typical AFM image of high-density SWCNTs for the construction of thin-film transistors. **g** Statistical distribution of the densities of different-chirality SWCNT films for the construction of thin-film transistors. Error bars in (**e**) and (**g**) are the standard deviation of statistics.

(SDS) to disperse SWCNTs (supplementary note 2 and supplementary Fig. 10). SDS is alkyl type surfactant, which has no selectivity towards the chiral angles of SWCNTs[28]. We constructed TFTs by the films of (7, 6), (8, 4), and (9, 2) SWCNTs, which belong to the same family (2*n* + *m* = 20 family). The transfer curves of the corresponding TFTs are shown in supplementary Fig. 11. The on-state current of the TFTs shows the same trend as that of the DOC-dispersed SWCNTs. These results further confirm that the measured chirality-dependent electrical properties are not dominated by the selective interaction of DOC with SWCNTs.

To quantitatively compare the electrical transport properties of different-chirality SWCNTs, we extracted the on-state current and mobility from two branches (*p*-branch and *n*-branch) of transfer curves following the conventional method (supplementary note 3 and supplementary Fig. 12)[35–39]. As shown in Fig. 3a-d, the general trend is that with an increase in diameter, both the on-state current and carrier mobility increase, which is consistent with previous reports including single-SWCNT FETs and TFTs[20,21,24]. Most importantly, the type and chiral angle of SWCNTs clearly exhibit a greater influence on their electrical transport properties, which has not yet been observed experimentally. For Type II SWCNTs, the on-state currents corresponding to both the *p*- and *n*-branches decrease with increasing chiral angle in the same family, while for Type I SWCNTs, the trend is the opposite except for the family of (7, 5) and (9, 1) SWCNTs, where the *p*-branch shows a deviant chiral angle dependence. Notably, for SWCNTs of the same family, the variation in the on-state current or carrier mobility with increasing chiral angle can be as high as one order

of magnitude. We also extracted threshold voltage ($V_{th}$) and hysteresis[36–38], and plotted their relationship with diameters (supplementary Fig. 13). There is no obvious type, family or chiral angle dependent relationship. However, it is notable that the hysteresis of (6, 5) and (10, 2) SWCNTs is clearly larger than other species, which is likely derived from their oxygen-doping defects caused by the oxide dielectric layer ($HfO_x$)[40,41]. Especially, (6, 5) SWCNTs have the largest hysteresis, because they are more easily oxidized due to their relatively larger curvature of C-C bonds[42]. The large hysteresis of the transistors has non-volatile characteristics, which has been demonstrated to be applicable in synaptic transistors for neuromorphic computing[43,44].

Different from ballistic transport of short-channel individual-SWCNT FETs[4], the TFTs constructed from thin films are driven by a more complex conduction mechanism[45,46]. As shown in Fig. 4a, in the conductive channel of a TFT, carriers are injected into SWCNTs from source electrodes, are transported in SWCNTs, jump over the junction between two overlapped SWCNTs and are finally collected by drain electrodes. In this process, the injection and collection of carriers should overcome the Schottky barrier between metal electrodes and semiconducting SWCNTs, which behaves as a contact resistance ($2R_c$ in Fig. 4a) in the macroscopic conduction behavior. Meanwhile, the as-injected carriers suffer from intratube transport (*SWCNT Resistance* ($R_{CNT}$) in Fig. 4a) in SWCNTs and intertube transport (*Junction* ($R_J$) in Fig. 4a) between two SWCNTs, which commonly behaves as a channel resistance ($R_L$).

To analyze the electrical transport properties of different SWCNTs in detail, we extracted $2R_c$ and $R_L$ from the transfer curves of

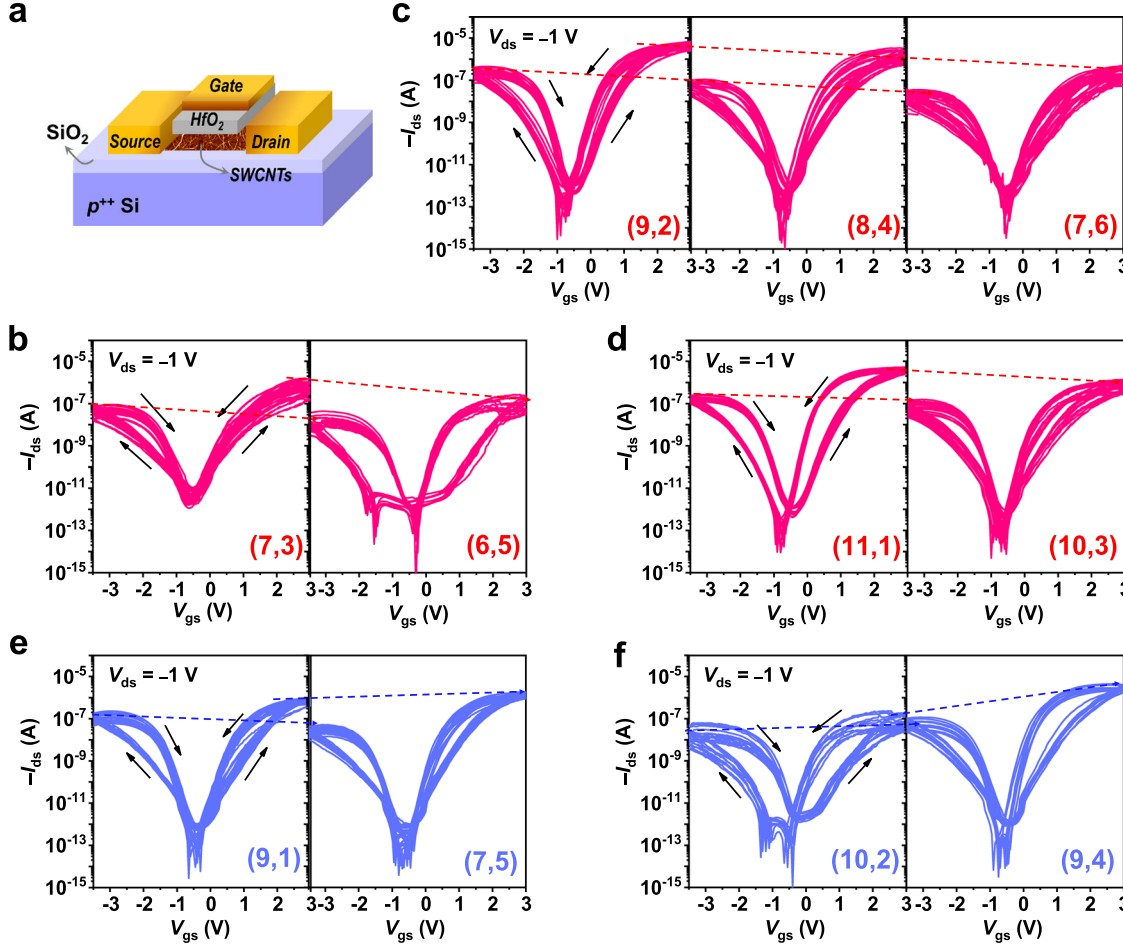

**Fig. 2 | Electrical characterization of transistors constructed from various single-chirality SWCNT films. a** Schematic diagram of a typical top-gate TFT. **b–f** Transfer curves of various single-chirality SWCNT TFTs, which were classified by their family (red for Type II SWCNTs and blue for Type I SWCNTs). Black arrows in (**b–f**) represent the forward and reverse scanning directions of the gate voltage $V_{gs}$. Dashed lines are used for the comparison of the on-state currents for the same-family SWCNTs.

SWCNT TFTs using a previously reported method (supplementary note 3)[36,38]. As shown in Fig. 4d, e, the contact resistance $2R_c$ shows significant type and chiral angle dependence, which is almost the same as that for the on-state current and mobility. Generally, the contact resistance comes from the Schottky barrier ($E_{SB}$), which is related to the difference between the Fermi level of metal electrodes and the top of the valence band (or bottom of the conduction band) of SWCNTs for holes (or electrons) (Fig. 4b). The corresponding Schottky barrier for holes $E_{SBh}$ and electrons $E_{SBe}$ can be described as[46]

$$E_{SBh} = E_F - E_v = \frac{1}{2}E_g - \varphi_M + \varphi_{center} \qquad (1)$$

$$E_{SBe} = E_c - E_F = \frac{1}{2}E_g + \varphi_M - \varphi_{center} \qquad (2)$$

where $E_{SBh}$ and $E_{SBe}$ are the Schottky barrier heights for holes and electrons, $E_F$, $E_v$ and $E_c$ are the Fermi levels of the metal electrode, valence band and conduction band of SWCNTs, respectively. $E_g$ is the bandgap of SWCNTs, and $\varphi_M$ and $\varphi_{center}$ are the affinity energy of the metal electrodes and center energy of the SWCNT bandgap, which are independent of chirality[25,46]. As an ideal quasi-one-dimensional material, the first bandgap ($S_{11}$) dominates the transport property of semiconducting SWCNTs[47,48]. When Ti/Pd is used as the drain/source electrodes with an identical fabrication process, the $\varphi_M$ values for all devices are the same, which means that the Schottky barrier height is

determined by the $S_{11}$ of SWCNTs. A smaller $S_{11}$ of SWCNTs results in a smaller Schottky barrier height between the SWCNTs and metal electrodes. The $S_{11}$ values of the eleven distinct single-chirality SWCNTs from the empirical Kataura plot[14] are shown in Fig. 4f. A comparison between $2R_c$ and $S_{11}$ indicates that, similar to $S_{11}$, the Schottky barrier height ($2R_c$) between Type I SWCNTs and electrodes decreases while that of Type II SWCNTs increases with increasing chiral angle in the same family, which fully supports that $S_{11}$ determines the barrier between SWCNTs and electrodes. In this way, the carriers are more easily injected from the source electrodes into the Type I SWCNTs with larger chiral angle. As a result, the SWCNTs exhibit higher on-state current and carrier mobility. In contrast, Type II SWCNTs with smaller chiral angles in the same family have higher on-state current and carrier mobility.

As mentioned above, the channel resistance ($R_L$) also plays a key role in the TFT performance. According to the $R_L$ extracted from the TFTs (Fig. 4g, h), the channel resistance obviously has a dependence on the type and chiral angle similar to the contact resistance. Type I SWCNTs in the same family with larger chiral angles exhibit lower resistance, while Type II SWCNTs with smaller chiral angles have decreasing resistance. The SWCNT networks in the channels of TFTs can be considered as two-dimensional random networks of conducting hollow cylinders[45,48,49]. As mentioned above, the channel resistances are contributed by nanotube segments and junctions. At room temperature, SWCNTs exhibit ballistic transport over distances to a micrometer[49–51]. In the present work, the average lengths of various ($n$,

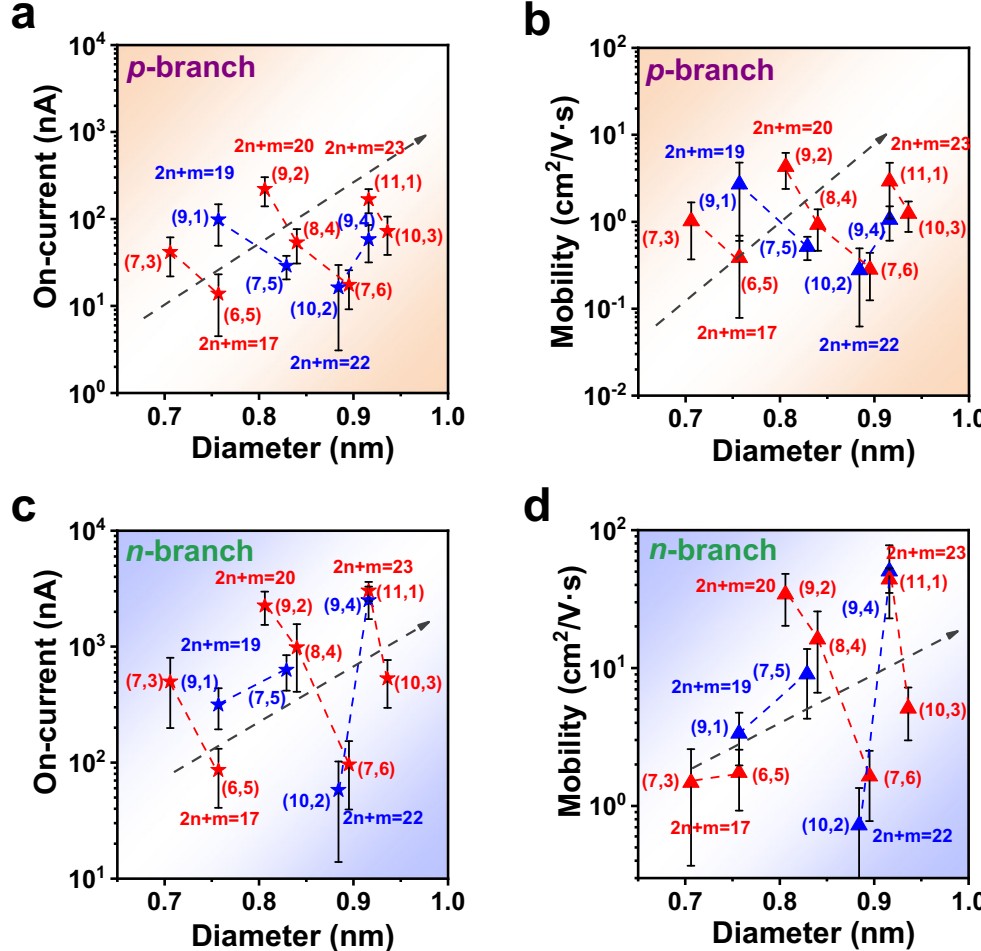

**Fig. 3 | On-state currents and mobility of the transistors constructed by different single-chirality SWCNT films as a function of their diameter.** **a**, **c** Statistical distribution of the on-state current; (**b**, **d**) statistical distribution of the mobility; (**a**, **b**) electrical features of the *p*-branch, (**c**, **d**) electrical features of the

*n*-branch. Blue and red symbols represent Type I and Type II SWCNTs, respectively. The black dashed arrows show the change trend of the on-state current and carrier mobility with increasing diameter. Error bars in the Figure are the standard deviation of statistics.

*m*) species (~230 nm) are much shorter than mean free paths[49–51]. We thus assume that the resistance of the contributing SWCNT segments can be neglected, and that the resistances of junctions are dominated in the channel resistances, which is demonstrated by experimental measurements[52–54]. With this assumption, the SWCNT networks are modeled to be a two-dimensional random junction resistor network, in which the charge transfer across junctions is considered as a hopping process[48,55]. Similar to the transport of carriers within the nanotubes, their transport across the junctions may be affected by intrinsic density of states (DOS) and surface potentials ($\varphi_s$)[45,48,49]. Thus, a conductance $G_{ij}^{NT}$ across a junction $ij$ connecting SWCNTs $i$ and $j$ can be described as refs. [48,56]:

$$G_{ij}^{NT}(E_F) = \iint_{E_{cb}-e\varphi_s}^{\infty} G_{ij}(E_F, E_i, E_j) \cdot D_i(E_i) \cdot F(E_i, E_F) \cdot D_j(E_j) \cdot F(E_j, E_F) dE_i dE_j$$

(3)

$$G_{ij}(E_F, E_i, E_j) = \frac{e^2 \cdot \omega_{ij}^{symm}}{4k_B T \cdot \cosh\left(\frac{E_i - E_F}{2k_B T}\right) \cdot \cosh\left(\frac{E_j - E_F}{2k_B T}\right)}$$

(4a)

$$\approx \frac{e^2 \omega_{ij}^{symm}}{k_B T} \exp\left(\frac{E_F}{k_B T} - \frac{E_i + E_j}{2k_B T}\right)$$

(4b)

where $E_F$ is the Fermi energy, $E_{cb}$ the energy of the bottom of conduction band, $e$ the elemental charge, $\varphi_s$ the surface potential created by gate voltage, $E_i$ and $E_j$ the energies in SWCNTs $i$ and $j$, respectively, $D_i(E)$ and $D_j(E)$ are their DOS, $F(E, E_F)$ the Fermi–Dirac distribution. $G_{ij}(E_F, E_i, E_j)$ the bond conductance of junction $ij$ at the energies of $E_i$ and $E_j$ for SWCNTs $i$ and $j$, $k_B$ the Boltzmann constant, $T$ the absolute temperature, $\omega_{ij}^{symm}$ a symmetrized hopping rate between SWCNTs $i$ and $j$. In equation (4),

$$\omega_{ij}^{symm} = \omega_0 \cdot \exp\left(-|\triangle E_{ij}|/2k_B T\right)$$

(5)

where $\omega_0$ a hopping prefactor determined by the charge-transfer integral between the states and $\triangle E_{ij}$ the energy difference ($E_j - E_i$) between the states $j$ and $i$. Due to the use of dense networks of single-chirality SWCNTs, many nanotube junctions are present and uniformly distributed in the channels from source to drain electrodes. We assume that $\triangle E_{ij}$ is the same for each junction. And thus, the term of $e^2 \omega_{ij}^{symm}/k_B T$ in Eq. 4b can be considered constant. The surface potential $\varphi_s$ increases with gate voltage, and approximately pinned near the conduction (or valance) band edges ($\sim E_g/2e$) in the on-regime of TFTs[57]. The theoretical simulation results show that the first conduction subband determines transport performance in practical devices[57,58]. Actually, only states with energies close to the band edge contribute to the transport due to the rapidly decaying of Fermi tail[58]. For simplicity, we only consider the first subband of the conduction

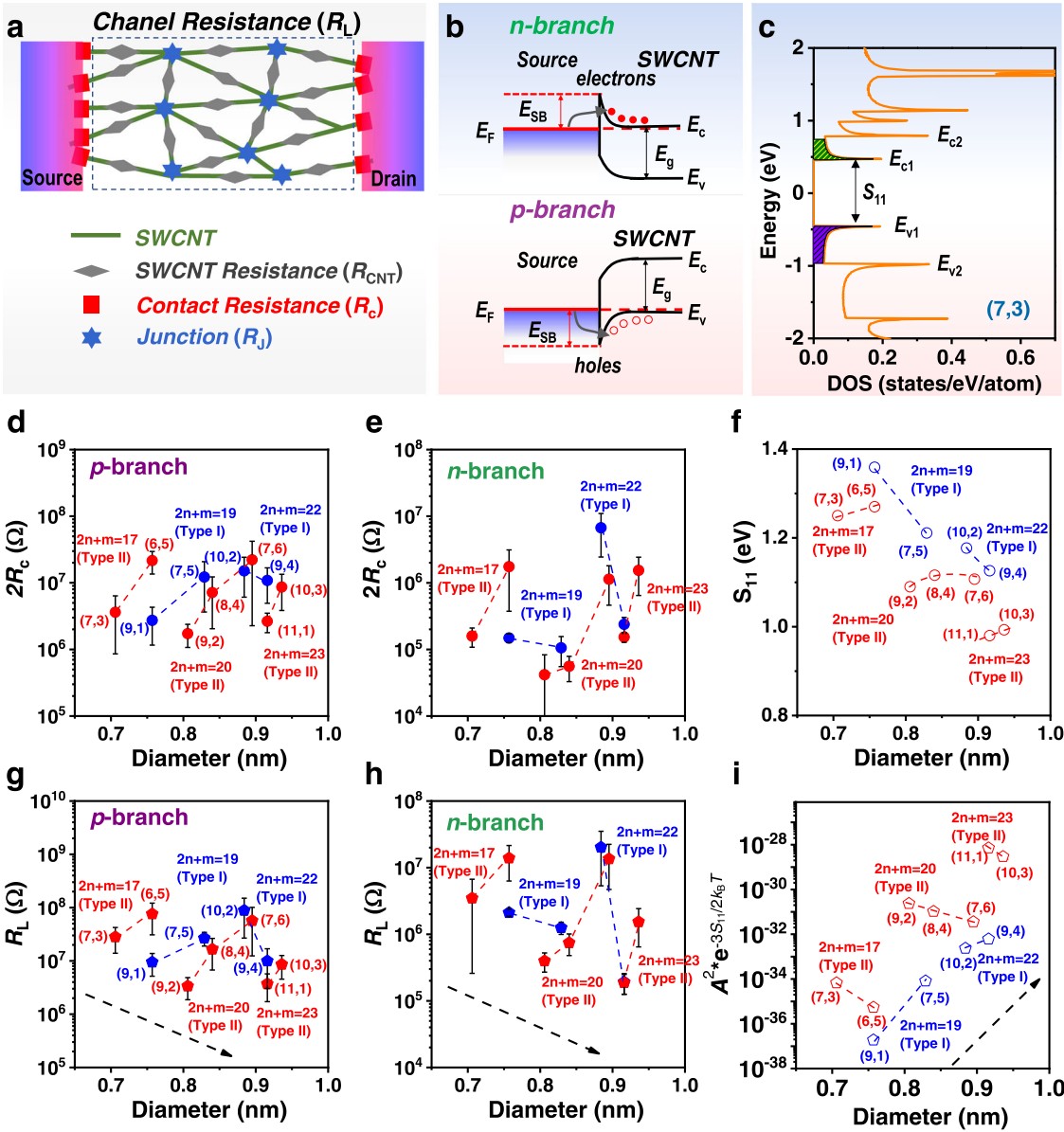

**Fig. 4 | Resistance analysis of TFTs constructed by different single-chirality SWCNTs. a** Schematic diagram of carrier transport from the source to drain electrode and composition of the contact resistance ($2R_c$) and channel resistance ($R_L = R_J + R_{CNT}$). **b** Schematic diagram of the construction of a Schottky barrier ($E_{SB}$) between an electrode and SWCNTs for electrons and holes. **c** Density of states (DOS) of (7, 3) SWCNTs[47], in which the first conduction and valence bands are indicated by the shaded areas. **d, e** Plots of the contact resistance ($2R_c$) of the $p$-branch (**d**) and $n$-branch (**e**) of SWCNT-TFTs as a function of their diameter. **f** The first bandgap ($S_{11}$ values) of the single-chirality species as a function of their

diameter, which were extracted from the Kataura plot in Ref. [14]. **g, h** Plots of the channel resistance ($R_L$) of the $p$-branch (**f**) and $n$-branch (**g**) of SWCNT-TFTs as a function of their diameter. **i** Plots of the relative junction conductance $G_{ij}^{NT'}$ as a function of SWCNT diameter deduced from Eq. (6) at room temperature ($k_B T = 26$ meV). Blue and red symbols represent Type I and Type II SWCNTs, respectively. Dashed lines are introduced as guides to the eye. The black dashed arrows show the change trend of the channel resistance or relative junction conductance with increasing diameter. Error bars in (**d**), (**e**), (**g**) and (**h**) are the standard deviation of statistics.

band. It has been demonstrated that the energy difference between the bottoms of the first and second conduction subbands is close to $E_g/2$[58]. With these assumptions, a conductance $G_{ij}^{NT}$ of a typical junction $ij$ in the on state can be calculated by plugging Eq. (4b) into Eq. (3) and integrating from $E_{cb} - e\varphi_s$ to $E_{cb} + E_g/2 - e\varphi_s$ in the first conduction band. Because the value of $\omega_0$ is unknown[48], the $G_{ij}^{NT}$ is further simplified to a relative junction conductance $G_{ij}^{NT'}$ as follows:

$$G_{ij}^{NT} \propto G_{ij}^{NT'} = A^2 \cdot e^{(-3S_{11})/2k_B T} \qquad (6)$$

where $A$ is the DOS area of the first conduction band and can be calculated by using the DOS distribution from a previous report (as

shown in supplementary Fig. 14)[47,48]. A typical DOS of an SWCNT such as (7, 3) is shown in Fig. 4c.

With Eq. (6), a relative junction conductance ($G_{ij}^{NT'}$) for different $(n, m)$ can be evaluated, and accordingly, the corresponding channel resistances could be compared. As shown in Fig. 4i, with increasing diameter, the relative junction conductance generally increases. In one family, with increasing chiral angle, the relative junction conductance increases for Type I SWCNTs, while it decreases for Type II SWCNTs. Clearly, the effects of the type and chiral angle of SWCNTs on junction resistance are opposite to their effects on the measured channel resistance, indicating that the resistance of SWCNT networks mainly derived from junction resistance. Additionally, the junction resistance

is also closely related to the overlap area roughly approximated as the product of the diameters between the intersecting tubes[49]. An increase in diameter thus decreases the junction resistance, which is consistent with the channel resistance decreasing with increasing diameter (Fig. 4g, h). According to Kausera's calculations[25,26], which considered both electron effective mass and phonon scattering by the ensemble Monte Carlo method and an iterative solution of Boltzmann's transport equation, chirality (type and chiral angle) has an effect on intrinsic carrier mobility similar to that on junction conductance. In other words, intrinsic resistance and mobility should have a similar dependence on their chiral structure. The above analysis reasonably explains the effect mechanism of the type and chiral angle of SWCNTs on the measured channel resistance, and thus the resulting on-current and mobility. To make it easier to understood, the relationship between the measured channel resistance and the relative junction conductance was plotted in supplementary Fig. 15. The channel resistance of the same-family SWCNTs decreases with an increase in junction conductance. For SWCNTs of different families, although the plotting of the channel resistance and relative junction conductance does not perfectly collapse down to a single curve, the general trend shows that it decreases with increasing junction conductance. This imperfect relationship may be caused by the influence of the carriers with higher energies in higher-order sub-bands[57,58] and structure-dependent defects produced in the ultrasonication dispersion of SWCNTs[59].

In general, based on the electronic band structure of different SWCNTs, we theoretically analyzed the influence of their type and chiral angle on the contact resistance between SWCNTs and electrodes, the intrinsic resistance of SWCNTs, and the intertube contact resistance, which explained our experimental results about the relationship between SWCNT chirality and the measured electrical properties. We also noted that the dependence of the electrical properties measured by experiments on their chiral structures was not completely consistent with the theoretical calculation results. For example, the electrical transport of (9, 1) and (7, 5) SWCNTs in p-type conduction is contrary to the theoretical prediction. The difference may be caused by surfactant residue on SWCNTs, impurities or the fluctuation of the interface state density[60]. With the removal of surfactants around SWCNTs, the further improvement of the chiral purity of SWCNTs and the reduction of the interface state density, the dependence of the electrical properties of SWCNTs on their chiral structures can be predicted to be more accurately established. Despite this, our current results provide an important reference for the design of high-performance SWCNT-based electronic and optoelectronic ICs.

In summary, we have experimentally explored how the chiral structures of SWCNTs affect their electrical transport properties by measuring the performance of TFTs constructed from 11 kinds of single-chirality SWCNT films. The results show that the electrical transport properties of SWCNTs are determined by their type and chiral angle. In the same family, Type I SWCNTs exhibit a higher on-state current and higher mobility with increasing chiral angle, while Type II SWCNTs show the opposite trend. According to theoretical calculations, the chirality-dependent electrical properties are derived from the electronic band structure, which determines the contact barrier height between metal electrodes and SWCNTs, the intrinsic resistance and the intertube contact resistance. These results give us a clear image of the relationship between the chirality and electrical properties of SWCNTs, providing important scientific guidance for the design of SWCNT-based electronic and optoelectronic devices.

## Methods
### Preparation of single-chirality SWCNT films
Single-chirality SWCNTs were separated by gel chromatography from HiPco-SWCNTs (diameter: $1.0 \pm 0.3$ nm, NanoIntegris)[28–30]. The SWCNT films were prepared by the alkaline small molecule regulation technique we developed[31]. Specifically, the surfactants in the as-prepared SWCNT solution were replaced with the single surfactant DOC at 0.05 wt% by using a centrifugal filter unit (Amicon Ultra-15 with an Ultracel-100K membrane, Merch Millipore) at a centrifugal force of $800 \times g$. The optical density of the SWCNT solution was controlled to 0.3 at 280 nm for all species. Before deposition, 0.05 M $NaHCO_3$ was introduced into the solution to enhance the interaction between SWCNTs and functionalized substrates. The $SiO_2$/Si substrates were ultrasonically cleaned in acetone and ethanol and subsequently immersed into PLL aqueous solution (0.1%, Sigma–Aldrich) for 15 min to achieve amine functionalization. Next, after washing with deionized water and drying with $N_2$ blowing, the amine-functionalized substrates were immersed into the as-prepared single-chirality SWCNT solution for 1 h to prepare films. Subsequently, the substrates were taken out and rinsed repeatedly with deionized water to remove the surfactants and dried by $N_2$ blowing for device fabrication. The low-density SWCNT films were prepared by decreasing the immersing time.

### TFT fabrication
The as-prepared SWCNT films were heat treated at 300 °C in vacuum for 1 h before device fabrication. Subsequently, top-gate TFTs with a channel length of 2 μm and width of 20 μm were constructed. The source and drain electrodes (Ti/Pd: 0.5/40 nm) of the TFTs were prepared by standard photolithography (MA6, Zeiss), a lift-off process and then thermal evaporation (TE, SKY Technology). The SWCNTs outside the channels were etched using reactive ion etching (RIE, Plasmalab 80 plus, Oxford Instruments). The dielectric was 15 nm $HfO_2$ deposited by an atomic layer deposition system (ALD, Savannah-100, Cambridge NanoTech). Finally, top gate electrodes (Ti/Pd: 0.5/40 nm) were deposited by thermal evaporation.

### Characterization
Optical absorption spectra were used to characterize the purity and concentration of SWCNT solutions with an ultraviolet-visible-infrared spectrophotometer (UV-3600, Shimadzu). Raman spectra were measured using a high-resolution confocal micro-Raman spectrometer (HR800, Horiba) equipped with three different excitation wavelengths of 488, 514, and 633 nm. The length of SWCNTs and the morphology and linear density of SWCNT films were characterized by AFM (Multi-Mode 8, Bruker). The performance of the TFTs was measured using a semiconductor analyzer (Keithley 4200) in a probe station (Lakeshore TTPX).

## Data availability
The key data generated in this study are provided in the Supplementary Information/Source Data file. Source data are provided with this paper.

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

## Acknowledgements

This work was financially supported by the National Key Research and Development Program of China (grant nos. 2020YFA0714700 and 2018YFA0208402), the National Natural Science Foundation of China (grant nos. 51820105002, 51872320, 51472264, 11634014, and 52172060), the Strategic Priority Research Program of the Chinese Academy of Sciences (grant no. XDB33030100), the Key Research Program of Frontier Sciences, CAS (grant no. QYZDBSSW-SYS028), and the Youth Innovation Promotion Association of CAS (grant no. 2020005).

## Author contributions

H.L. proposed and supervised the project. W.S. fabricated most of TFT devices and measured their electrical properties. X.L. performed Raman spectra and TEM characterization, and fabricated partial TFT devices, and measured their electrical performances. L.L. and D.Y performed the separation experiment of single-chirality SWCNTs. F.W., X.W., and W.Z. provided technical support on device fabrication, electrical measurement, and Raman spectra characterization. W.S. drafted the manuscript. H.L. reviewed and revised the final manuscript. H.K. and S.X. were involved in the discussions. All authors analyzed and discussed the experimental data.

## Competing interests

The authors declare no competing interests.
