## [Peer Review File · Nature Communications]

REVIEWER COMMENTS

Reviewer #1 (Remarks to the Author):

Previous papers have reported on how the electrical properties of carbon nanotubes vary in field effect transistors as a function of nanotube diameter, showing increasing current with increasing diameter.

This paper makes a nice attempt at going one step further and experimentally measuring the behaviors of nanotubes in field effect transistors as a function of chiral angle and chiral index. The paper positively combines state of the art nanotube sorting techniques with device measurements. A better understanding of how nanotube transport properties vary with chiral angle and chiral index would be important to the field. However, there are major issues in the paper that need to be addressed before the paper can be better evaluated.

1) One of the conclusions of the paper is that “The differences in the electrical properties of SWCNTs with different chirality can be attributed to their different electronic band structures, which determine the contact barrier between electrodes and SWCNTs, intrinsic resistance and intertube contact resistance.” However:

a) The paper analyzes on-current, mobility, contact resistance, and channel resistance all versus diameter. If bandgap and density of states are the most important factor like the authors claim, then plots of on-current, mobility, contact resistance, and channel resistance versus band gap or versus $A \cdot \exp(-\text{gap}/kT)$ should be provided that show a clear relationship that collapses to something near a single curve, independent of n,m family or chiral angle or diameter.

b) Roughly guessing what the above curve will look like, it does not seem that there will be a nice relationship between $A \cdot \exp(-\text{gap}/kT)$ and on-current, mobility, contact resistance, or channel resistance. For example, $A \cdot \exp(-\text{gap}/kT)$ changes by 3 orders of magnitude when going from the $2n+m=23$ family to the $2n+m=17$ family but the $2R_c$ does not significantly change and the channel resistance changes by only 1 order of magnitude. Thus, it is not clear what is causing the observed dependencies.

2) The adhesion strength of the insulating sorter molecule DOC to the sidewalls of the SWNTs could significantly vary with chiral angle and diameter. That is presumably in part how the purification by (n,m) works. The completeness of the removal of DOC could then vary somewhat strongly with the

(n,m), diameter, and chiral angle – thereby affecting the electrical properties (both contact resistance and channel resistance). How do we know that the electrical data presented here are not dominated by this effect? For example, is DOC sticking to type-II nanotubes with increasing chiral angle, more strongly?

3) Why do some n,m have more hysteresis than others? Is this complicating interpretation of the data?

4) The $2R_c$ should be subtracted off before calculating mobility.

5) Please provide more plots of threshold voltage and hysteresis versus chirality or family type or diameter.

Reviewer #2 (Remarks to the Author):

The manuscript by Liu et al report the investigation of single chirality SWCNTs solutions and try to extract information on the chirality dependent difference in transport properties.

My main concerns about this work is the design of the experimental part.

- The authors prepare single chirality solutions by chromatography, but they investigate the properties of the nanotube in a network. Now the network may behave in a slightly different way depending on the local density of SWNTs, which may be prone to experimental optimization. So, if the authors did not optimize, the concentration the deposition technique I'm afraid that the samples may not be fully comparable, in particular since they decided to go for sparse networks. Also, would be important to describe how the SWNTs are deposited.

- Furthermore, why they claim that samples are single chirality and they show absorption, this is not a sufficient characterization to exclude the presence of a very limited number of metallic tubes and chirality different than the main one.

Finally, the results are interesting, but the authors should clarify and verify their hypothesis before this work can be accepted for publication.

Reviewer #3 (Remarks to the Author):

This paper addresses the chiral index dependence of single-walled carbon nanotubes as an alternative to silicon for electronic and optoelectronic circuits. This will become increasingly important as the energy consumption of information and communication technologies continues to increase: already 200 TW hours/year for data centres alone, and expected to more than double by 2030. Therefore the possibility of using materials other than silicon is of global importance.

The information in the paper could be provided in a more logical order. For example, Type I and Type II are referred to on Page 4, but not explained until Page 6.

What is the significance of the hysteresis in Fig. 2? Could this provide the basis of memristor-like behaviour, for example for low-energy neuromorphic computing?

Notwithstanding such recommendations and questions, the paper seems to be based on robust experiments by the authors which they relate rigorous theory in the literature.

I recommend publication in Nature Communications.

Response to Reviewers:

Reviewer #1:

Previous papers have reported on how the electrical properties of carbon nanotubes vary in field effect transistors as a function of nanotube diameter, showing increasing current with increasing diameter.

This paper makes a nice attempt at going one step further and experimentally measuring the behaviors of nanotubes in field effect transistors as a function of chiral angle and chiral index. The paper positively combines state of the art nanotube sorting techniques with device measurements. A better understanding of how nanotube transport properties vary with chiral angle and chiral index would be important to the field. However, there are major issues in the paper that need to be addressed before the paper can be better evaluated.

Reply: Thank you for your positive comments and valuable recommendations about our work. According to your suggestions, we have made the corresponding modifications to our manuscript as shown in the following.

Comments 1-1. One of the conclusions of the paper is that “The differences in the electrical properties of SWCNTs with different chirality can be attributed to their different electronic band structures, which determine the contact barrier between electrodes and SWCNTs, intrinsic resistance and inter-tube contact resistance.”

However:

a) The paper analyzes on-current, mobility, contact resistance, and channel resistance all versus diameter. If bandgap and density of states are the most important factor like the authors claim, then plots of on-current, mobility, contact resistance, and channel resistance versus band gap or versus $A \cdot \exp(-\text{gap}/kT)$ should be provided that show a clear relationship that collapses to something near a single curve, independent of n,m family or chiral angle or diameter.

b) Roughly guessing what the above curve will look like, it does not seem that there will be a nice relationship between $A \cdot \exp(-\text{gap}/kT)$ and on-current, mobility, contact

resistance, or channel resistance. For example, $A \cdot \exp(-\text{gap}/kT)$ changes by 3 orders of magnitude when going from the $2n+m=23$ family to the $2n+m=17$ family but the $2R_c$ does not significantly change and the channel resistance changes by only 1 order of magnitude. Thus, it is not clear what is causing the observed dependencies.

Reply 1-1: Thank you for your valuable comments and suggestions. As you suggested, we have plotted the relationship between on-current, mobility, contact resistance, and channel resistance versus $A \cdot \exp(-\text{gap}/kT)$, as shown in the following Figure R1. Similar to the SWCNT diameter, the general trend is that the on-state current and carrier mobility of the transistors increases with increasing $A \cdot \exp(-\text{gap}/kT)$ while $2R_c$ and R_L exhibit opposite trend for both n and p branches. The different is that, for the same family, on-current, mobility, contact resistance, and channel resistance versus $A \cdot \exp(-\text{gap}/kT)$ change linearly with increasing $A \cdot \exp(-\text{gap}/kT)$, indicating that the electronic band structures play an important role in their transport performances for the same-family SWCNTs. To make the description more rigorous, we modify the conclusion you mentioned above.

Figure R1 The relationship between $A \cdot \exp(-\text{gap}/kT)$ and on-current, mobility, contact resistance, and channel resistance.

Previous: Page 2, line 14

“The differences in the electrical properties of SWCNTs with different chiralities can

be attributed to their different electronic band structures, which determine the contact barrier between electrodes and SWCNTs, intrinsic resistance and intertube contact resistance.”

Revised: Page 2, line 14

The differences in the electrical properties of **the same-family** SWCNTs with different chiralities can be attributed to their different electronic band structures, which determine the contact barrier between electrodes and SWCNTs, intrinsic resistance and intertube contact resistance.

Previous: (None)

Revised: Page 14, Line 2

To make it easier to understand, the relationship between the measured channel resistance and carrier concentration was plotted in Figure S12. It is clear that the channel resistance of the same-family SWCNTs decreases with increasing their carrier concentration.

Previous: (None)

Revised: Supporting Information 12, Page 15

The relationship between the channel resistance and carrier concentration.

中国科学院物理研究所

Institute of Physics CAS

Figure S12. The relationship between the channel resistance and the carrier concentration $A \cdot \exp(-\text{gap}/kT)$. The results show that the channel resistance of the same-family SWCNTs decreases with increasing the carrier concentration.

Comments 1-2. The adhesion strength of the insulating sorter molecule DOC to the sidewalls of the SWNTs could significantly vary with chiral angle and diameter. That is presumably in part how the purification by (n, m) works. The completeness of the removal of DOC could then vary somewhat strongly with the (n, m) , diameter, and chiral angle – thereby affecting the electrical properties (both contact resistance and channel resistance). How do we know that the electrical data presented here are not dominated by this effect? For example, is DOC sticking to type-II nanotubes with increasing chiral angle, more strongly?

Reply 2: Thank you for your valuable comments. To reduce the effect of DOC adsorption on the electrical properties of SWCNTs, the concentration of DOC in SWCNT solution was tuned to 0.05 wt %, which is the lowest concentration that can stably disperse SWCNTs. After film deposition, the surfactant molecules were removed by repeated cleaning with deionized water. TEM image show that few DOC molecules are coated on the SWCNTs (Figure S6). These results indicate that most of the DOC surfactants have been removed from the as-prepared SWCNT films. And the differences in the electrical transport performances of distinct (n, m) SWCNT films should not be dominated by the adhesion strength of DOC on different SWCNTs.

To exclude the effect of the adhesion strength of DOC to SWCNTs on their electrical properties, we employed achiral molecules SDS to disperse SWCNTs and deposited the SWCNT films. SDS is alkyl type surfactant, which has no selectivity towards the chiral angles of SWCNTs. We constructed TFTs by Type II SWCNTs films of $(7, 6)$, $(8, 4)$, and $(9, 2)$ SWCNTs, which belong to the same family ($2n + m = 20$ family). The resulted transfer curves of the corresponding TFTs are shown in **Figure S8**. The on-state current of the TFTs shows the same trend as that of the DOC-dispersed SWCNTs. These results further confirm that the chirality-dependent

electrical properties are not dominated by the selective interaction of DOC with SWCNTs.

Previous: (none)

Revised:

Page 6, line 26

To reduce the effect of the surfactant adsorbed on SWCNTs on their electrical properties, sodium deoxycholate (DOC) with a concentration of 0.05% was employed to disperse various (n, m) SWCNTs, which is the lowest concentration that can stably disperse SWCNTs. After film deposition, the surfactant molecules were removed by repeated cleaning with deionized water. The results show that few DOC molecules remained on SWCNTs (Figure S6).

Previous: (none)

Revised:

Page 8, line 12

To exclude the effect of the adhesion strength of DOC to SWCNTs on their electrical properties, we employed achiral molecules sodium sulfate (SDS) to disperse SWCNTs. SDS is alkyl type surfactant, which has no selectivity towards the chiral angles of SWCNTs.²⁸ We constructed TFTs by Type II SWCNTs films of $(7, 6)$, $(8, 4)$, and $(9, 2)$ SWCNTs, which belong to the same family ($2n + m = 20$ family). The transfer curves of the corresponding TFTs are shown in Figure S8. The on-state current of the TFTs shows the same trend as that of the DOC-dispersed SWCNTs. These results further confirm that the measured chirality-dependent electrical properties are not dominated by the selective interaction of DOC with SWCNTs.

Previous: (none)

Revised: Supporting Information 6, page 7

6. High-resolution TEM image of a single SWCNT in a deposited SWCNT film.

Figure S6. A typical TEM image of a single SWCNT from a deposited film. Before deposition, the SWCNTs were dispersed in a DOC solution with a concentration of 0.05%. After film deposition, the surfactant molecules were removed by repeated cleaning with deionized water. The results show that few DOC molecules remained on SWCNTs.

Previous: (none)

Revised: Supporting Information, Page 9

8. The performances of the TFTs constructed by SDS-dispersed SWCNTs.

Figure S8. The transfer curves of the TFTs constructed by Type II SWCNTs films of (7, 6), (8, 4) and (9, 2) SWCNTs, which were deposited by SDS-dispersed solutions. The on-state current of the TFTs shows the same trend as that of the DOC-dispersed

SWCNTs. These results further confirm that the measured chirality-dependent electrical properties are not dominated by the selective interaction of DOC with SWCNTs.

Comments 1-3. Why do some (n, m) have more hysteresis than others? Is this complicating interpretation of the data?

Reply 3: Thank you for your interesting questions. To explore the hysteresis of (n, m) SWCNTs, we plotted the relationship of hysteresis versus diameter. As shown in Figure S10, except for the larger hysteresis of the TFTs constructed by $(6, 5)$ and $(10, 2)$ SWCNTs, the hysteresis of the other (n, m) SWCNTs is not very different. In other words, the hysteresis has no obvious dependence on the diameters, chiral angle, type and family of SWCNTs. The hysteresis in SWCNT-TFTs usually arises from defects, including: interface defects such as hydroxyl groups ($-OH$) between SWCNTs and the oxide gate dielectric layer and bulk defects within dielectric layer (Dai Hongjie, *et al. Nano Lett.*, **2003**, 3, 193; Stephen A. McGill, *et al. Appl. Phys. Lett.*, **2006**, 89, 163123; John A. Rogers, *et al. Adv. Funct. Mater.*, **2012**, 22, 2276-2284). Charging and discharging of defects by carriers in the channel region electrostatically modulate the conductivity of SWCNTs, producing the hysteresis. The larger hysteresis of $(6, 5)$ SWCNTs is likely derived from the oxygen-doping of $(6, 5)$ by the oxide dielectric layer ($HfOx$) (Stephen K. Doorn, *et al., Adv. Funct. Mater.* 2015, 25, 6157–6164; Jana Zaumseil, *Adv. Optical Mater.* 2022, 10, 2101576). Compared with other species, the $(6, 5)$ SWCNTs is more easily oxidized due to their relatively larger curvature of C-C bonds (Wei Xiaojun, *et al., J. Phys. Chem. C* 2016, 120, 10705–10710).

Previous: none

Revised: Page 9, line 6

We also extracted threshold voltage (V_{th}) and hysteresis,³⁶⁻³⁸ and plotted their relationship with diameters (Figure S10). There is no obvious type, family or chiral angle dependent relationship for both threshold voltage and hysteresis. However, it is

notable that the hysteresis of (6, 5) and (10, 2) SWCNTs is clearly larger than other species, which is likely derived from their oxygen-doping defects caused by the oxide dielectric layer (HfO_x).^{40, 41} Especially, (6, 5) SWCNTs have the largest hysteresis, because they are more easily oxidized due to their relatively larger curvature of C-C bonds.⁴² The large hysteresis of the transistors has non-volatile characteristics, which has been demonstrated to be applicable in synaptic transistors for neuromorphic computing.^{43,44}

Comments 1-4. The $2R_c$ should be subtracted off before calculating mobility.

Reply 4: Thank you for your suggestions. Following your constructive suggestion, we extracted the mobility through YFM in which the $2R_c$ has been subtracted off before calculating mobility, and the results were shown in **Figure 3**.

Previous: Page 9, Figure 3

Figure 3. The on-state currents and mobility of the transistors constructed by different single-chirality SWCNT films as a function of their diameters. (a, c) the statistical distribution of on-current; (b, d) the statistical distribution of mobility; (a, b) The electrical features of p -branch, (c, d) the electrical features of n -branch. Blue and red symbols represent Type I and Type II SWCNTs, respectively. The black dotted arrows show the change trend of the on-state current and carrier mobility with an increase in diameters.

Revised: Page 9, Figure 3

Figure 3. The on-state currents and mobility of the transistors constructed by different single-chirality SWCNT films as a function of their diameters. (a, c) the statistical distribution of on-current; (b, d) the statistical distribution of mobility; (a, b) The electrical features of p -branch, (c, d) the electrical features of n -branch. Blue and red symbols represent Type I and Type II SWCNTs, respectively. The black dotted arrows show the change trend of the on-state current and carrier mobility with an increase in diameters.

red symbols represent Type I and Type II SWCNTs, respectively. The black dotted arrows show the change trend of the on-state current and carrier mobility with an increase in diameters.

Supporting Information

Previous:

Page 7, line 1

6. Methods to extract threshold voltage, on-current and mobility.

(1) **Threshold voltage (V_{th}).** The V_{th} is obtained from YFM curve where Y is defined as:^{3,4}

$$Y = \frac{I_{ds}}{\sqrt{g_m}}$$

where g_m is the transconductance. In the YFM curve, the extension line of linear segment intersects with V_{gs} axis and the intersection is V_{th} .

Figure S6. A typical YFM curve calculated from transfer curve of a TFT.

(2) **On-current.** On-current is defined as the current at $V_{gs} = V_{th} \pm 1$ V (+ for n -branch and - for p -branch) in the transfer characteristics of I_{ds} - V_{gs} .³

(3) **Mobility calculation.** The mobility of TFT is extracted by following equation:⁴

$$u = \frac{L}{V_{ds} C_{ox} W} \cdot \frac{dI_{ds}}{dV_{gs}} = \frac{L}{V_{ds} C_{ox}} \cdot \frac{g_m}{W}$$

where L and W are channel length and width respectively, V_{gs} and V_{ds} are the gate voltage and drain voltage respectively, and C_{ox} is the gate capacitance. In our calculation, L , W , V_{ds} are 2 μm , 20 μm , and -1 V respectively. g_m is the maximum transconductance which is extracted by taking the derivative of the transfer characteristics curve. The gate capacitance C_{ox} per unit area is calculated by the following equation, which considers the electrostatic coupling between SWCNTs:

$$C_{ox} = \left\{ C_Q^{-1} + \frac{1}{2\pi\epsilon_0\epsilon_{ox}} \ln \left[\frac{\Lambda_0 \sinh(2\pi t_{ox}/\Lambda_0)}{R} \right] \right\}^{-1} \Lambda_0^{-1}$$

where $C_Q = 4.0 \times 10^{-10} \text{ F/m}$ is the quantum capacitance of nanotubes, $\epsilon_0\epsilon_{ox} = 25 \times 8.85 \times 10^{-14} \text{ F/cm}$ is the dielectric constant at the interface between nanotubes film and HfO_2 , R is the average radius of every single chirality SWCNT, t_{ox} is the thickness of HfO_2 layer which is 15 nm, and Λ_0^{-1} is the linear density of SWCNT film which has been shown in **Figure 1** in main text.

7. Methods to extract contact resistance ($2R_c$) and channel resistance (R_L).

According to previous reports,^{3,4} contact resistance and channel resistance can be obtained by YFM. The total resistance of TFT can be expressed as

$$R_{\text{tot}} = \frac{V_{ds}}{I_{ds}} = 2R_c + R_L$$

where R_L can be approximated as

$$R_L = \frac{L}{\mu * W * C_{ox}(V_{gs} - V_{th})}$$

In YFM, Y function is defined as

$$Y = \frac{I_{ds}}{\sqrt{g_m}} = \sqrt{\frac{W}{L} \mu C_{ox} V_{ds} (V_{gs} - V_{th})}$$

just like shown in **Figure S6**. Therefore, if we define the slope of linear segment of YFM curve as a parameter α , then

$$\alpha = \sqrt{\frac{W}{L} \mu C_{\text{ox}} V_{\text{ds}}}$$

So, R_L can be described as

$$R_L = \frac{V_{\text{ds}}}{\alpha^2 (V_{\text{gs}} - V_{\text{th}})}$$

Meanwhile, $2R_c$ can be described as

$$2R_c = \frac{V_{\text{ds}}}{I_{\text{ds}}} - \frac{V_{\text{ds}}}{\alpha^2 (V_{\text{gs}} - V_{\text{th}})}$$

where α and V_{th} are both obtained from linear segment of YFM curve.

Revised:

Page 9, line 1

9. Methods to extract threshold voltage, on-current, mobility, contact resistance ($2R_c$) and channel resistance (R_L).

(1) **Threshold voltage (V_{th}).** The V_{th} is obtained from YFM curve where Y is defined as:^{3,4}

$$Y = \frac{I_{\text{ds}}}{\sqrt{g_m}}$$

where g_m is the transconductance. In the YFM curve, the extension line of linear segment intersects with V_{gs} axis and the intersection is V_{th} .

Figure S9. A typical YFM curve calculated from transfer curve of a TFT.

(2) **On-current.** On-current is defined as the current at $V_{gs} = V_{th} \pm 1$ V (+ for n -branch and - for p -branch) in the transfer characteristics of I_{ds} - V_{gs} .³

(3) **Hysteresis.** Hysteresis is defined as the difference of the gate voltage at $I_{ds} = 1$ nA in forward and reverse scanning transfer curves.

(4) **Mobility, $2R_c$ and R_L calculation.** The total resistance of TFT can be expressed as

$$R_{tot} = \frac{V_{ds}}{I_{ds}} = 2R_c + R_L$$

where R_L can be approximated as

$$R_L = \frac{L}{\mu * W * C_{ox}(V_{gs} - V_{th})}$$

Obviously, u is mobility excluding $2R_c$. Therefore,

$$\frac{V_{ds}}{I_{ds}} = 2R_c + R_L = \frac{L}{\mu * W * C_{ox}(V_{gs} - V_{th})} + 2R_c$$

$$\text{and } I_{ds} = \frac{V_{ds} * u * W * (V_{gs} - V_{th})}{L + 2R_c * u * W * C_{ox}(V_{gs} - V_{th})}$$

Because of $2R_c$ almost has no relationship with V_{gs} , therefore

$$g_m = \frac{dI_{ds}}{dV_{gs}} = \frac{V_{ds} * u * W / L}{\left[1 + 2R_c * W * u * \frac{C_{ox}}{L} * (V_{gs} - V_{th})\right]^2}$$

In YFM, Y function is defined as

$$Y = \frac{I_{ds}}{\sqrt{g_m}} = \sqrt{\frac{W}{L} \mu C_{ox} V_{ds} (V_{gs} - V_{th})}$$

just like shown in **Figure S9**. Therefore, if we define the slope of linear segment of YFM curve as a parameter α , then

$$\alpha = \sqrt{\frac{W}{L} \mu C_{ox} V_{ds}}$$

So, mobility can be described as

$$u = \frac{L}{V_{ds} C_{ox}} * \frac{\alpha^2}{W}$$

where L and W are channel length and width respectively, V_{gs} and V_{ds} are the gate voltage and drain voltage respectively, and C_{ox} is the gate capacitance. In our calculation, L , W , V_{ds} are 2 μm , 20 μm , and -1 V respectively. The gate capacitance C_{ox} per unit area is calculated by the following equation, which considers the electrostatic coupling between SWCNTs:

$$C_{ox} = \left\{ C_Q^{-1} + \frac{1}{2\pi\epsilon_0\epsilon_{ox}} \ln \left[\frac{A_0 \sinh(2\pi t_{ox}/A_0)}{R\pi} \right] \right\}^{-1} A_0^{-1}$$

And, R_L can be described as

$$R_L = \frac{V_{ds}}{\alpha^2(V_{gs} - V_{th})}$$

Meanwhile, $2R_c$ can be described as

$$2R_c = \frac{V_{ds}}{I_{ds}} - \frac{V_{ds}}{\alpha^2(V_{gs} - V_{th})}$$

where α and V_{th} are both obtained from linear segment of YFM curve.

Comments 1-5. Please provide more plots of threshold voltage and hysteresis versus chirality or family type or diameter.

Reply 5: Thank you for your valuable suggestions, we plotted the relationship of threshold voltage and hysteresis versus diameter, as shown in Figure S9. There is no obvious structure dependent threshold voltage and hysteresis for both p-branch and n-branch.

Previous: none

Revised: Page 9, line 6

We also extracted threshold voltage (V_{th}) and hysteresis,³⁶⁻³⁸ and plotted their relationship with diameters (Figure S10). There is no obvious type, family or chiral angle dependent relationship for both threshold voltage and hysteresis. However, it is notable that the hysteresis of (6, 5) and (10, 2) SWCNTs is clearly larger than other species, which is likely derived from their oxygen-doping defects caused by the oxide

dielectric layer (HfO_x).^{40, 41} Especially, (6, 5) SWCNTs have the largest hysteresis, because they are more easily oxidized due to their relatively larger curvature of C-C bonds.⁴² The large hysteresis of the transistors has non-volatile characteristics, which has been demonstrated to be applicable in synaptic transistors for neuromorphic computing.^{43,44}

Previous: (none)

Revised: Supporting Information, Page13, line1

10. The relationship of threshold voltage (V_{th}) and hysteresis of (n, m) SWCNTs versus their diameters.

Figure S10. The relationship of threshold voltage and hysteresis with chirality, family, Type and diameter. The results show there is no obvious type, family or chiral angle dependent relationship for both threshold voltage and hysteresis. However, it is notable that the hysteresis of (6, 5) and (10, 2) SWCNTs is clearly larger than other

中国科学院物理研究所

Institute of Physics CAS

species, which is likely derived from their oxygen-doping defects caused by the oxide dielectric layer (HfO_x).

Reviewer #2:

Comments 2-1: The manuscript by Liu *et al.* report the investigation of single chirality SWCNTs solutions and try to extract information on the chirality dependent difference in transport properties. The authors prepare single chirality solutions by chromatography, but they investigate the properties of the nanotube in a network. Now the network may behave in a slightly different way depending on the local density of SWNTs, which may be prone to experimental optimization. So, if the authors did not optimize, the concentration the deposition technique I'm afraid that the samples may not be fully comparable, in particular since they decided to go for sparse networks. Also, would be important to describe how the SWNTs are deposited.

Reply 2-1: Thank you for positive and valuable comments. Indeed, we optimized the deposition process and density of SWCNTs.

To reduce the performance difference caused by the SWCNT density, we prepared highly uniform SWCNT films with average linear density of 32 ± 5 tubes/ μm for constructing thin film transistors (TFTs) by the alkaline small molecule regulation technique we developed previously (Su Wei, *et al.* Carbon, 2020, 163, 370-378). With this technique, the density difference in the SWCNTs films with linear density higher than 30 tubes/ μm has been proven to be less than 8% in a 4-inch wafer, indicating that the films have excellent uniformity. In the present work, the transfer curves of TFTs for each (n, m) species almost overlap, sufficiently confirming that the measured electrical properties of distinct (n, m) SWCNT films are fully comparable.

In the method part, we describe the deposition process of SWCNT high-density and high-uniform films in detail. Specifically, the concentration of DOC surfactant dispersed SWCNTs are tuned to be 0.05 wt%. The concentration of SWCNTs was controlled by their optical density of 0.3 at 280 nm for all species. Before deposition,

the 0.05 M NaHCO₃ was introduced into solution to enhance the interaction between SWCNTs and functionalized substrates. The SiO₂/Si substrates were ultrasonically cleaned in acetone and ethanol, and subsequently immersed into PLL aqueous solution (0.1%, Sigma-Aldrich) for 15 min to achieve amine-functionalization. Next, after washing with deionized water and drying with N₂ blowing, the amine-functionalized substrates were immersed into the as-prepared single-chirality SWCNT solution for 1 h to prepare films. As a result, 11 kinds of SWCNT films have a linear density with 32±5 tubes/μm (Figures 1f, g and S5). In other words, the difference in the transport properties of distinct (*n*, *m*) species are not dominated by the SWCNT density differences.

With regard to the sparse network in the present work, we employ it for the characterization of the length distribution of the (*n*, *m*) SWCNTs. The linear density of the sparse network is less than ~5 tubes/μm, which were immersed in SWCNT solution for about 1 min. For comparison, we also fabricated (6, 5) TFTs using low-density SWCNT films with linear densities of ~20 and ~10 nanotubes/um. As shown in Figure S4, the performance and uniformity of TFTs are both poor because of percolation path and linear density variation. Therefore, the electrical properties of different (*n*, *m*) species are not comparable by the low-density films.

Page 5, Figure 1d and g captions

Previous:

(d) Typical atomic force microscopy (AFM) image of a low-density SWCNT film. (e) Statistical distribution of the lengths of different-chirality SWCNTs. (f) Typical AFM image of high-density SWCNTs for the construction of thin-film transistors. (g) Statistical distribution of the densities of different-chirality SWCNT films.

Revised:

(d) Typical atomic force microscopy (AFM) image of a low-density SWCNT film **for length measurement**. (e) Statistical distribution of the lengths of different-chirality

SWCNTs. (f) Typical AFM image of high-density SWCNTs for the construction of thin-film transistors. (g) Statistical distribution of the densities of different-chirality SWCNT films **for the construction of thin-film transistors.**

Previous: Page 6, line 22

Figures 1f and S5 shows a typical image of the SWCNT films used to construct TFTs, whose linear densities were controlled at approximately 32 ± 5 tubes/ μm (**Figure 1g**).

Revised: Page 6, line 22

It is notable that, low-density SWCNT films is not suitable for the comparison of the electrical properties of different (n, m) species due to their poor electrical performances and uniformity (Figure S4). Figures 1f and S5 shows a typical image of the **high-density** SWCNT films used to construct TFTs, whose linear densities were controlled at approximately 32 ± 5 tubes/ μm (**Figure 1g**). **Previous:**

Previous: Page 15, Line 23

Subsequently, the substrates were removed, rinsed repeatedly with deionized water and dried by N_2 blowing for device fabrication.

Revised:

Page 16, line 16

Subsequently, the substrates were **taken out and** rinsed repeatedly with deionized water **to remove the surfactants** and dried by N_2 blowing for device fabrication. **The low-density SWCNT films were prepared by decreasing the immersing time.**

Previous: None.

Revised: Supporting Information 4

4. Transfer curves of the TFTs constructed by the low-density (6,5) SWCNTs.

Figure S4. The performances of the TFTs constructed by low-density (6, 5) SWCNT films. (a) and (b) show the AFM images of (6,5) SWCNT films with linear density of ~ 10 and ~ 20 tubes/ μm . (c) and (d) is the corresponding transfer curves of the TFTs. The detailed fabrication process of TFTs is described in the method part in the main text. The fabricated transistors are top-gate structures with channel length and width of 2 and 20 μm , respectively, in which 15-nm-thick HfO_2 was used as a dielectric layer.

Comments 2-2. why they claim that samples are single chirality and they show absorption, this is not a sufficient characterization to exclude the presence of a very limited number of metallic tubes and chirality different than the main one.

Reply 2: Thank you for your questions.

Optical absorption spectroscopy is one of the simplest, rapidest and most accurate methods to characterize the structure distribution of SWCNTs. Due to the different band structures, different SWCNTs exhibit different optical absorption peaks

corresponding to the first and second subbands. Through the characterization of the optical absorption spectra of the separated SWCNTs, their chiral purity can be evaluated based on the peak area ratio of the optical absorption peaks derived from different species (Liu, H.; et al., Nat. Commun. 2011, 2, 309). Therefore, in the present work, we characterized the structure purity of the single-chirality SWCNTs by optical absorption spectroscopy. As shown in Figures 1a and S1, the chiral purities of most of the species are higher than 85%. In particular, (6, 5), (7, 3), (7, 5) and (8, 4) SWCNTs exhibit a chiral purity higher than 90%. Due to the high chiral purity of single-chirality SWCNTs used in the present work, the experientially measured electrical properties of each (n, m) SWCNT film should mainly reflect their own electrical properties.

Raman spectroscopy is another powerful method to characterize the structure distribution of SWCNTs, which is very sensitive to the detection of trace amounts of metallic SWCNTs and different chiral species under resonant excitation. However, the characterization of different (n, m) species require different resonance excitation wavelengths, and it is difficult to accurately characterize the chiral purity of SWCNTs by limited excitation wavelengths. As you suggested, we further measured the Raman spectra of 11 types of single-chirality species with 488, 514 and 633 nm of excitation wavelengths, and detect whether trace amounts of metallic SWCNTs with different diameters are present in these SWCNT films. The results show that no signals of metallic SWCNTs are detected with these three excitation wavelengths, indicating that the effect of metallic SWCNTs on the measured electrical properties of each (n, m) species should be negligible.

Previous: (none)

Revised:

Page 6, line 6

Metallic SWCNTs in each (n, m) species were detected by Raman spectra with excitation wavelengths of 488, 514 and 633 nm. And the results show that no signals

of metallic SWCNTs are detected (Figure S2).

Previous: (none)

Revised:

Page 16, line 9

Raman spectra were measured using a high-resolution confocal micro-Raman spectrometer (HR800, Horiba) equipped with three different excitation wavelengths of 488, 514 and 633 nm.

Supporting Information 2

Previous: (none)

Revised:

Page 3, line 1

2. Raman spectra of the single-chirality SWCNTs used in the present work.

Figure S2. Raman spectra of the single-chirality SWCNTs used in the present work with excitation wavelengths of 488, 514 and 633 nm, respectively. As a comparison, Raman spectra of the Raw HiPco-SWCNTs were also measured. The results show that no metallic SWCNTs were detected in the single-chirality species.

Reviewer #3:

Comments:

This paper addresses the chiral index dependence of single-walled carbon nanotubes as an alternative to silicon for electronic and optoelectronic circuits. This will become increasingly important as the energy consumption of information and communication

中国科学院物理研究所

Institute of Physics CAS

technologies continues to increase: already 200 TW hours/year for data centres alone, and expected to more than double by 2030. Therefore, the possibility of using materials other than silicon is of global importance.

The information in the paper could be provided in a more logical order. For example, Type I and Type II are referred to on Page 4, but not explained until Page 6.

What is the significance of the hysteresis in Fig. 2? Could this provide the basis of memristor-like behavior, for example for low-energy neuromorphic computing?

Notwithstanding such recommendations and questions, the paper seems to be based on robust experiments by the authors which they relate rigorous theory in the literature.

I recommend publication in Nature Communications.

Reply: Thank you for your positive comments and valuable recommendation. According to the issues you raised, we have made corresponding revisions to our manuscript, which are highlighted in red color.

Comments 3-1. The information in the paper could be provided in a more logical order. For example, Type I and Type II are referred to on Page 4, but not explained until Page 6.

Reply 3-1: Thank you for your constructive suggestions. According to your suggestions, we have made corresponding revisions, which are highlighted in red color.

Previous:

Page 4, line 3

However, due to the limited types and yields of single-chirality SWCNTs that can be prepared, the dependence between the chiral structure and electrical transport properties of SWCNTs has not been reported yet.

Revised:

Page 4, line 3

Except for diameter, semiconducting SWCNTs can be divided into Type I and Type II according to $\text{mod}(2n + m, 3) = 1$ or 2. Meanwhile, each type of SWCNT can be divided into different families based on the value of $(2n+m)$. The SWCNTs with the same value of $(2n + m)$ belong to the same family, in which with increasing diameter, the chiral angle increases.²⁷ However, due to the limited types and yields of single-chirality SWCNTs that can be prepared, the dependence between the chiral structure and electrical transport properties of SWCNTs has not yet been reported.

Previous:

Page 6, line 10

According to the above classification rules, the 11 kinds of single-chirality SWCNTs used in our study contain both types and five families of species (as shown in **Figure 1c**).

Revised:

Page 6, line 10

According to the above classification rules **of type and family**, the 11 kinds of single-chirality SWCNTs used in our study contain both types and five families of species (as shown in **Figure 1c**).

Comments 3-2: What is the significance of the hysteresis in Fig. 2? Could this provide the basis of memristor-like behavior, for example for low-energy neuromorphic computing?

Reply 3-2: Thank you for your interesting question. Generally speaking, the hysteresis in FETs derives from the charge-trapping caused by the defects at the interface between the channel and the gate dielectric and within the gate dielectric. With a positive bias on the gate, carriers are trapped by defects while changing to a negative bias on the gate, carriers will gradually release. Therefore, the transistors with large hysteresis have non-volatile characteristics, which has been demonstrated to be applicable in synaptic transistors for neuromorphic computing (Ping Feng, *et al.*

Adv. Funct. Mater. **2017**, 1604447; Esqueda I. S. *et al.*, *ACS Nano* 2018, 12, 7352; Wan H. *et al.*, *ACS Nano* 2020, 14, 10402; Zhao J. *et al.*, *Nano Res.* 2021, 14, 4258;).

Previous: (none)

Revised:

Page 9, line 6

We also extracted threshold voltage (V_{th}) and hysteresis,³⁶⁻³⁸ and plotted their relationship with diameters (Figure S10). There is no obvious type, family or chiral angle dependent relationship for both threshold voltage and hysteresis. However, it is notable that the hysteresis of (6, 5) and (10, 2) SWCNTs is clearly larger than other species, which is likely derived from their oxygen-doping defects caused by the oxide dielectric layer (HfO_x).^{40, 41} Especially, (6, 5) SWCNTs have the largest hysteresis, because they are more easily oxidized due to their relatively larger curvature of C-C bonds.⁴² The large hysteresis of the transistors has non-volatile characteristics, which has been demonstrated to be applicable in synaptic transistors for neuromorphic computing.^{43,44}

REVIEWER COMMENTS

Reviewer #1 (Remarks to the Author):

I want to like this paper but am having a hard time knowing if its main conclusions are widely applicable to all nanotubes or just the nanotubes in this study.

1) Within the same $2n+m$ family, the current and mobility decrease with increasing chiral angle. This is true for 4 of the 5 families measured. This is a very interesting observation. But, it is not true for the 5th family. 4 out of 5 seems like it *might* be an actual trend but maybe not. It is interesting but not conclusive.

2) The newly included data plotting current and mobility versus exponential of the bandgap (Fig. R1) does not nicely collapse the data down to anything close to a single curve. There is barely any correlation. So, the bandgap / barrier heights do not seem to be the most important factor. Then what is?

3) The above cast doubt on whether the observed differences in FET behaviors are *intrinsic* properties of nanotubes or not. There could be multiple extrinsic factors.

4) One possible extrinsic factor is DOC surfactant remaining on the nanotube surfaces. Newly included TEM data were added to the paper to prove that the nanotubes are DOC free. However, such a surfactant coating is not necessarily stable in the TEM. More newly included data have been added to the paper in which SDS surfactant were use to show that DOC surfactant is not the culprit. However, it is not clear if SDS can displace DOC surfactant, which is strongly bound? Could the DOC still be on the nanotubes nonetheless?

5) Another possible extrinsic factor is point defects (e.g., oxidation) on the nanotube sidewalls. Even at defect densities below that detectable by Raman, electrical transport properties can be significant affected. The paper may simply be measuring the susceptibility of various chiralities of nanotubes to have point defects or to acquire them during processing. Alternatively, the gel chromatography may be selecting different n,m according to the density of defects. Thus, the paper may simply be reporting how defective their particular nanotubes are when processed in their particular way.

6) Other:

A) When the paper says “carrier concentration” do the authors really mean “density of states”? The carrier concentration in the on-state will be set by the gate voltage and gate capacitance which will be n,m -independent.

B) Percolation threshold. The on-current in a percolating network will change super linearly and very rapidly with nanotube density, if the density is near the percolation threshold. Small differences in nanotube density could therefore translate to large differences in on-current. This effect would need to be ruled out by calculating the percolation threshold (nanotubes per area) of each n,m network (which will be length-dependent) and analyzing how close to percolation each network is. The tubes with 20 nanotube per micrometer included in the SI show almost no on-current suggesting 30 nanotubes per micrometer could be near the threshold.

C) Are the heights of the nanotubes in the networks the same? If not, then there may be many more nanotubes per area in some networks than others.

Reviewer #2 (Remarks to the Author):

The authors have clarified most of the critical points raised by the reviewers. The manuscript can be accepted in the present form.

Reviewer #3 (Remarks to the Author):

I previously recommended publication, and my recommendation has not changed.

Response to Reviewers:

Reviewer #1:

I want to like this paper but am having a hard time knowing if its main conclusions are widely applicable to all nanotubes or just the nanotubes in this study.

Reply: Thank you for your positive comments and valuable recommendations about our work. We believe that our main conclusions are widely applicable to all nanotubes. We addressed the issues you raised point by point in the following.

Comments 1-1. Within the same $2n+m$ family, the current and mobility decrease with increasing chiral angle. This is true for 4 of the 5 families measured. This is a very interesting observation. But, it is not true for the 5th family. 4 out of 5 seems like it *might* be an actual trend but maybe not. It is interesting but not conclusive.

Reply 1-1: Thank you for your comments. Although five families of SWCNTs were employed to investigate the structure dependence of their electrical properties, the performances of both p- and n-branches of the corresponding transistors were explored. The results show that nine of the ten data sets show that the on-current and mobility of SWCNTs exhibit a strong chiral angle dependence, indicating that the results are widely applicable to SWCNTs. The abnormal trend of the electrical transport between (9, 1) and (7, 5) SWCNTs in p-type conduction is not clear yet, which may be affected by the residual surfactant molecules on the nanotube surfaces or the fluctuations of the interface state density in the dielectric layer (*Appl. Phys. Lett.* 2018, 113, 083105). We added the corresponding discussion in the revised manuscript.

Previous: Page 14, Lines 22-28

For example, the electrical transport of (9, 1) and (7, 5) SWCNTs in p-type conduction is contrary to the theoretical prediction. The difference may be caused by surfactant residue on SWCNTs and impurities. With the removal of surfactants around

SWCNTs and the further improvement of the chiral purity of SWCNTs, the dependence of the electrical properties of SWCNTs on their chiral structures can be predicted to be more accurately established.

Revised: Page 14, Lines 22-28

For example, the electrical transport of (9, 1) and (7, 5) SWCNTs in p-type conduction is contrary to the theoretical prediction. The difference may be caused by surfactant residue on SWCNTs, impurities or **the fluctuation of the interface state density**⁵². With the removal of surfactants around SWCNTs, the further improvement of the chiral purity of SWCNTs and **the reduction of the interface state density**, the dependence of the electrical properties of SWCNTs on their chiral structures can be predicted to be more accurately established.

Comments 1-2. The newly included data plotting current and mobility versus exponential of the bandgap (Fig. R1) does not nicely collapse the data down to anything close to a single curve. There is barely any correlation. So, the bandgap / barrier heights do not seem to be the most important factor. Then what is?

Reply 1-2: Thank you for your kind comments. As demonstrated in the manuscript, for the same family of SWCNTs, their electrical transport performances are linearly related to their first sub-band, indicating that the electronic band structures play an important role in their transport performances for the same-family SWCNTs. For SWCNTs of different families, although the relationship between the on-state current versus exponential of the first sub-band does not perfectly collapse down to a single curve, the general trend shows that it increases with increasing carrier concentration. This imperfect relationship may be caused by the influence of the carriers in higher-order sub-bands such as the second sub-band [J. Appl. Phys., 2008, 104, 064515; IEEE Trans. Electron Devices, 2008, 55, 289]. However, it is difficult to determine the influence weights of the higher-order sub-bands for different-structure SWCNTs on their electrical transport performances due to their complexity, which is

still in progress. In our present conclusion, we emphasize the variation of the electrical properties of SWCNTs in the same family. We added the corresponding discussion in the revised manuscript.

Previous: Page 14, Lines 6-14 in the main manuscript

To make it easier to understood, the relationship between the measured channel resistance and carrier concentration was plotted in **Figure S15**. It is clear that the channel resistance of the same-family SWCNTs decreases with increasing their carrier concentration.

Revised: Page 14, Lines 6-14, in the revised manuscript

To make it easier to understood, the relationship between the measured channel resistance and carrier concentration was plotted in **Figure S15**. It is clear that the channel resistance of the same-family SWCNTs decreases with increasing their carrier concentration. **For SWCNTs of different families, although the plotting of the channel resistance and the carrier concentration in the first sub-band does not perfectly collapse down to a single curve, the general trend shows that it decreases with increasing carrier concentration. This imperfect relationship may be caused by the influence of the carriers in higher-order sub-bands^{50,51}.**

Comments 1-3. The above cast doubt on whether the observed differences in FET behaviors are *intrinsic* properties of nanotubes or not. There could be multiple extrinsic factors.

Reply 1-3: Thank you for your thought-provoking questions. We have carefully examined and studied the suggestions you mentioned as follows.

Comments 1-4. One possible extrinsic factor is DOC surfactant remaining on the nanotube surfaces. Newly included TEM data were added to the paper to prove that the nanotubes are DOC free. However, such a surfactant coating is not necessarily stable in the TEM. More newly included data have been added to the paper in which

中国科学院物理研究所

Institute of Physics CAS

SDS surfactant were used to show that DOC surfactant is not the culprit. However, it is not clear if SDS can displace DOC surfactant, which is strongly bound? Could the DOC still be on the nanotubes nonetheless?

Reply 1-4: Thank you for your comments. We agree with you that a few DOC molecules remain on SWCNT surfaces. However, we also measured the electrical properties of the SDS-dispersed SWCNTs (Figure S11). The results show a consistent trend. To verify whether DOC can be displaced by SDS, we characterized the optical absorption spectra of the DOC-dispersed (6, 5) SWCNTs before and after being displaced by SDS. The optical absorption spectra of the DOC and SDS-dispersed SWCNTs are different due to different dielectric environment. As shown in Figure S10, the S_{11} optical absorption peak of the DOC-dispersed (6, 5) SWCNTs exhibit a blue-shift of 4 nm after replacement with SDS, which coincided with that of the (6, 5) SWCNTs directly dispersed by SDS, sufficiently indicating that the DOC coating was replaced by SDS. We added the corresponding characterization results and discussion in the revised manuscript and Supplementary Information.

Previous: Page 7, Line 4 in the manuscript

The results show that few DOC molecules remained on SWCNTs (Figure S8).

Revised: Page 7, Line 4 in the manuscript

The results show that a few DOC molecules remained on SWCNTs (Figure S8).

Previous: None

Revised: Page 11, Supplementary Information 10 and Figure S10 in the revised Supplementary Information

The optical absorption spectra of the DOC and SDS-dispersed SWCNTs are different due to different dielectric environment. To verify whether DOC can be displaced by SDS, we characterized the optical absorption spectra of the DOC-dispersed (6, 5) SWCNTs before and after being displaced by SDS. As shown in

Figure S10, the S_{11} optical absorption peak of the DOC-dispersed (6, 5) SWCNTs exhibit a blue-shift of 4 nm after replacement with SDS, which coincided with that of the SDS-dispersed (6, 5) SWCNTs (black curve), sufficiently indicating that the DOC coating was replaced by SDS.

Figure S10. The optical absorption spectra of the DOC-dispersed (6, 5) SWCNTs before and after displacing with SDS surfactant. After displacing with SDS, the optical absorption spectrum exhibits a blue shift of 4 nm, which coincide with that of the SDS-dispersed (6, 5) SWCNTs. The results indicate that the DOC coating around SWCNTs could be displaced with SDS. For comparison, the optical absorption spectrum of the DOC dispersed (6,5) SWCNTs is upshifted vertically.

Comments 1-5. Another possible extrinsic factor is point defects (e.g., oxidation) on the nanotube sidewalls. Even at defect densities below that detectable by Raman, electrical transport properties can be significant affected. The paper may simply be measuring the susceptibility of various chiralities of nanotubes to have point defects or to acquire them during processing. Alternatively, the gel chromatography may be selecting different n,m according to the density of defects. Thus, the paper may

中国科学院物理研究所

Institute of Physics CAS

simply be reporting how defective their particular nanotubes are when processed in their particular way.

Reply 1-5: Thank you for your suggestions. The separation of single-chirality SWCNTs has been demonstrated to be caused by the different surfactant densities instead of the difference in the density of point defects on their sidewalls [*Nat. Commun.* 2011, 2, 309; *Sci. Adv.* 2021, 7, eabe0084, *Carbon*, 2022, 195, 349]. The single-chirality species employed in this work were separated from the same raw materials, which were dispersed and separated in the same process. For the construction of transistors, the film deposition and device fabrication process were also the same. Therefore, the single-chirality species employed in this work should have the similar defect density on their sidewalls, and defects should not be the main reason that induced the difference in the TFT behaviors of different (n, m) SWCNTs.

Comments 1-6: Other:

A) When the paper says “carrier concentration” do the authors really mean “density of states”? The carrier concentration in the on-state will be set by the gate voltage and gate capacitance which will be n, m -independent.

Reply 1-6A: Thank you for your comments. In the manuscript, the intrinsic carrier concentration (N_i) is the total number of occupied states in the subbands and can be calculated by integrating density of state in conductance band (for electron) and the Fermi Dirac distribution [*IEEE Trans. Electron Devices* 2008, 55, 289-297], which is strongly dependent on the chiral structures of SWCNTs.

$$N_i = \int_{E_{cl}}^{E_{c1}} DOS(E)f(E)dE$$

Considering the applied gate voltage in a classical MOS structure, the carrier concentration in the on-state can be described as [Liu Enke, et al., *The physics of semiconductors* (7th Edition), Publishing house of electronics industry, 2013]:

$$N \propto N_i * e^{|\varphi_s|/2kT}$$

Where k is Boltzmann constant, T is temperature, and φ_s is surface potential in

中国科学院物理研究所

Institute of Physics CAS

semiconductor which is determined by applied gate voltage and gate capacitance. In our present work, the on-state currents of different transistors were measured under the same gate voltage and gate capacitance. Therefore, the difference in the carrier concentration of distinct (n, m) SWCNTs in the on-state is determined by that of intrinsic carrier concentration. In other words, the carrier concentration of SWCNTs in the on-state is (n, m) -dependent under the same gate voltage.

B) Percolation threshold. The on-current in a percolating network will change super linearly and very rapidly with nanotube density, if the density is near the percolation threshold. Small differences in nanotube density could therefore translate to large differences in on-current. This effect would need to be ruled out by calculating the percolation threshold (nanotubes per area) of each n,m network (which will be length-dependent) and analyzing how close to percolation each network is. The tubes with 20 nanotube per micrometer included in the SI show almost no on-current suggesting 30 nanotubes per micrometer could be near the threshold.

Reply 1-6B: Thank you for your suggestions. As you suggested, we calculated the percolation threshold of each kind of (n, m) SWCNT film based on the following equation [*Nano Lett.* 2004, 4, 2513-2517]:

$$N_{th} = \frac{1}{\pi} \left(\frac{4.236}{L_{cnt}} \right)^2$$

Where N_{th} is the percolation threshold of (n, m) SWCNTs, L_{cnt} represents mean length of (n, m) SWCNTs. The results show that the linear percolation thresholds of (n, m) SWCNTs employed in this work is about 10 tubes/ μm (as shown in Table S1 and Figure S6), which is much lower than the average linear density of the SWCNT films (~ 32 tubes/ μm) used to construct transistors in our present work (Figure 1g). Therefore, small fluctuation in the average densities of different chiral SWCNTs would not change the difference in TFT behaviors.

On the other hand, the SWCNT films with an average density of ~ 32 tubes/ μm show high density uniformity across the whole substrate for different (n, m) species

(as shown in Figure S7), resulting in the high uniformity and repeatability in the electrical performances of the as-fabricated transistors (Figure 2). The corresponding transfer curves of the transistors almost overlap, which provides an important basis for studying the dependence of the electrical properties of SWCNTs on their chiral structures. In contrast, when the linear density of the films was reduced to less than ~20 tubes/ μm (Figure S4), the performance uniformity of the corresponding transistors is too poor to be used to explore the structure-dependent electrical performances of SWCNTs due to their worse density uniformity in 40- μm^2 channel of FETs. We added the corresponding discussion and characterization results in the revised manuscript and revised Supplementary Information:

Previous: Page 6 Lines 24-26, in the manuscript

Figures 1f and S5 shows a typical image of the high-density SWCNT films used to construct TFTs, whose linear densities were controlled at approximately 32 ± 5 tubes/ μm (Figure 1g).

Revised: Page 6 Lines 24-29, in the revised manuscript

Figures 1f and S5 shows a typical image of the high-density SWCNT films used to construct TFTs, whose linear densities were controlled at approximately 32 ± 5 tubes/ μm (Figure 1g), which is much higher than the percolation threshold (~10 tubes/ μm) (Table S1 and Figure S6). And the films of different (n, m) species show high uniformity across the substrates (Figure S7).

Previous: None

Revised: Page 7 in the Supplementary Information

6. The calculation of the percolation threshold.

The percolation threshold of each kind of (n, m) SWCNT film was calculated based on the following equation:^{2, 3}

$$N_{th} = \frac{1}{\pi} \left(\frac{4.236}{L_{cnt}} \right)^2$$

Where N_{th} is the percolation threshold of (n, m) SWCNTs, L_{cnt} represents mean length of (n, m) SWCNTs. The results show that the percolation thresholds of (n, m) SWCNTs is about 10 tubes/ μm (as shown in Table S1 and Figure S6), which is much lower than the average linear density of the SWCNT films (~ 32 tubes/ μm) used to construct transistors in our present work (Figure 1g). Therefore, small fluctuation in the average densities of different chiral SWCNTs would not change the difference in TFT behaviors.

Table S1 The calculated percolation threshold.

(n, m)	(7, 3)	(6, 5)	(9, 2)	(8, 4)	(7, 6)	(11, 1)	(10, 3)	(9, 1)	(7, 5)	(10, 2)	(9, 4)
Length (nm)	238.70	222.40	269.60	210.60	206.10	210.00	218.20	231.40	222.60	225.70	223.10
Area Density (tubes/ μm^2)	100.24	115.48	78.58	128.78	134.46	129.52	119.96	106.67	115.27	112.12	114.75
Linear density (tubes/ μm)	10.01	10.75	8.86	11.35	11.60	11.38	10.95	10.33	10.74	10.59	10.71

Figure S6. The calculated percolation threshold of different (n, m) SWCNTs based on their average lengths.

Previous: None

Revised: Page 8 in the supplementary Information

7. The characterization of the heights of the nanotubes in the networks

Figure S7. Height characterization of the nanotubes in the networks by AFM. The (n, m) SWCNT films with an average density of ~ 32 tubes/ μm show high density uniformity.

C) Are the heights of the nanotubes in the networks the same? If not, then there may be many more nanotubes per area in some networks than others.

Reply 1-6C: Thank you for your suggestions. As you suggested, we characterized the heights of the nanotubes in the 11 kinds of (n, m) SWCNT films by AFM. As shown in Figure S7, the height distribution of the SWCNTs in the 11 kinds of (n, m) SWCNT films is almost the same. We added the corresponding discussion and characterization results in the revised manuscript and revised Supplementary Information.

Previous: Page 6 Lines 24-26, in the manuscript

Figures 1f and S5 shows a typical image of the high-density SWCNT films used to construct TFTs, whose linear densities were controlled at approximately 32 ± 5

tubes/ μm (Figure 1g).

Revised: Page 6 Lines 24-29, in the revised manuscript

Figures 1f and S5 shows a typical image of the high-density SWCNT films used to construct TFTs, whose linear densities were controlled at approximately 32 ± 5 tubes/ μm (Figure 1g), which is much higher than the percolation threshold (~ 10 tubes/ μm) (Table S1 and Figure S6). And the films of different (n, m) species show high uniformity across the substrates (Figure S7).

Previous: None

Revised: Page 8 in the supplementary Information

7. The characterization of the heights of the nanotubes in the networks

Figure S7. Height characterization of the nanotubes in the networks by AFM. The (n, m) SWCNT films with an average density of ~ 32 tubes/ μm show high density uniformity.

Reviewer #2:

The authors have clarified most of the critical points raised by the reviewers. The manuscript can be accepted in the present form.

Reply: Thank you for your positive comments and valuable recommendation.

Reviewer #3:

I previously recommended publication, and my recommendation has not changed.

Reply: Thank you for your positive comments and valuable recommendation.

REVIEWER COMMENTS

Reviewer #1 (Remarks to the Author):

I still want to like this paper. The experimental data are really interesting. But, the explanation behind the data still do not make enough sense. Even if the explanation is not perfect, the paper can be published. Others can potentially follow up on the results and make more sense of it and advance the field. With this said, the paper still needs to address the following:

1) Regarding defects. The response letter states, “the single-chirality species employed in this work should have the similar defect density on their sidewalls, and defects should not be the main reason that induced the difference in the TFT behaviors of different (n, m) SWCNTs.”

However, the reactivity of SWCNTs is indeed n,m dependent. Even if the sorting is not defect dependent, the formation of defects during ultrasonication may be (n,m) dependent. See an example of n,m dependent chemistry here doi 10.1002/smll.201202761.

The authors also refer to this preferential reactivity in their statement about 6,5 being “more easily oxidized due to their relatively larger curvature of C-C bonds.⁴²”

In most cases the nanotubes with larger chiral angle have worse conductivity in this paper.

2) Carrier concentration. The response letter states “the intrinsic carrier concentration (N_i) is the total number of occupied states in the subbands and can be calculated by integrating density of state in conductance band (for electron)”.

This statement helps clarify that the intrinsic carrier concentration dependences are being analyzed in the main text and Fig. S15.

The question is then, why is the intrinsic carrier concentration of relevance at all in this paper? The experimental measurements are in on-state with carrier concentration set by gate and not related to intrinsic concentration.

Issues that have been fixed:

3) Surfactant exchange. The new data showing spectral shift after replacement of DOC with SDS are a nice addition to the paper and convincingly show that surfactant exchange has happened.

4) Percolation. The length and percolation measurements are good additions. Convincing that percolation behaviors not responsible for observed trends.

5) Diameter. The new diameter data are good additions. Convincingly show there is no major difference.

Response to Reviewers:

Reviewer #1:

I still want to like this paper. The experimental data are really interesting. But, the explanation behind the data still do not make enough sense. Even if the explanation is not perfect, the paper can be published. Others can potentially follow up on the results and make more sense of it and advance the field. With this said, the paper still needs to address the following:

Reply: Thank you for your positive comments and valuable recommendations.

Comments 1-1. Regarding defects. The response letter states, “the single-chirality species employed in this work should have the similar defect density on their sidewalls, and defects should not be the main reason that induced the difference in the TFT behaviors of different (n, m) SWCNTs.” However, the reactivity of SWCNTs is indeed (n, m) dependent. Even if the sorting is not defect dependent, the formation of defects during ultrasonication may be (n, m) dependent. See an example of (n, m) dependent chemistry here doi 10.1002/sml.201202761. The authors also refer to this preferential reactivity in their statement about (6, 5) being “more easily oxidized due to their relatively larger curvature of C-C bonds.⁴²” In most cases the nanotubes with larger chiral angle have worse conductivity in this paper.

Reply 1-1: Thank you for your comments. We agree with you that the (n, m)-dependent defects may be generated during ultrasonication due to the different reactivity of distinct (n, m) SWCNTs, which is possibly below detectable by Raman. However, the reactivity reported in the literature is dependent on the diameter and chiral angle of carbon nanotubes, and none of business for Type [Small 2013, 9, 8, 1379].

In comparison, the measured electrical preperformances of SWCNTs in the present work are strongly dependent on their Type and chiral angles. With increasing chiral angle for the same-family SWCNTs, Type I SWCNTs exhibit increasing

on-state current and mobility, while Type II SWCNTs show the reverse trend, which indicate that defects should not be the main reason that induced the difference in the TFT behaviors of different (n, m) SWCNTs. From this viewpoint, the defects produced in ultrasonication process are likely one of the reasons that caused the imperfect relationship between the on-state current versus exponential of the first sub-band. We added the corresponding discussion into the revised manuscript.

Previous: Page 14, Lines 25-26, in the manuscript

This imperfect relationship may be caused by the influence of the carriers in higher-order sub-bands^{50, 51}.

Revised: Page 14, Lines 25-27, in the revised manuscript

This imperfect relationship may be caused by the influence of the carriers in higher-order sub-bands^{51, 52} and **structure-dependent defects produced in the ultrasonication dispersion of SWCNTs**⁵³.

Comments 1-2. Carrier concentration. The response letter states “the intrinsic carrier concentration (N_i) is the total number of occupied states in the subbands and can be calculated by integrating density of state in conductance band (for electron)”. This statement helps clarify that the intrinsic carrier concentration dependences are being analyzed in the main text and Fig. S15. The question is then, why is the intrinsic carrier concentration of relevance at all in this paper? The experimental measurements are in on-state with carrier concentration set by gate and not related to intrinsic concentration.

Reply 1-2: Thank you for your comments. As stated in the previous response letter, under an applied gate voltage in a classical MOS structure, the carrier concentration (N) in the on-state can be described as [Liu Enke, et al., The physics of semiconductors (7th Edition), Publishing house of electronics industry, 2013]:

$$N \propto N_i * e^{|\varphi_s|/2kT}$$

Where N_i is the intrinsic carrier concentration, k is Boltzmann constant, T is temperature, and φ_s is surface potential in semiconductor which is determined by applied gate voltage and gate capacitance. In our present work, due to the same transistor structure and applied gate voltage, the surface potential φ_s are identical for different (n, m) SWCNTs. Based on equation 1, the difference in the carrier concentration of distinct (n, m) SWCNTs in the on-state is determined by that of intrinsic carrier concentration. Therefore, in main manuscript, we employ the intrinsic carrier concentration of different (n, m) SWCNTs to explain the differences in their electrical properties. In order to make it easier to understand, the corresponding discussion was added in the revised manuscript.

Previous: Page 13, Line 3-7 in the manuscript

The intrinsic resistance of SWCNTs is closely related to the carrier concentration and phonon scattering. For the typical density of states (DOS) of an SWCNT such as (7, 3), which is shown in **Figure 4c**, the intrinsic carrier concentration dominated by the first conduction band area can be described by⁴⁸

Revised: Page 13, Lines 3-17 in the revised manuscript

The intrinsic resistance of SWCNTs is closely related to the carrier concentration and phonon scattering. **The on-state carrier concentration (N) under an applied gate voltage can be described as⁴⁹:**

$$N \propto N_i * e^{|\varphi_s|/2kT} \quad (3)$$

Where N_i is the intrinsic carrier concentration, k is Boltzmann constant, T is temperature, and φ_s is surface potential in semiconductors which is determined by applied gate voltage and gate capacitance. Due to the same transistor structure and applied gate voltage, the φ_s is identical for distinct (n, m) SWCNTs. Therefore, the difference in intrinsic resistance is mainly determined by the intrinsic carrier concentration. For simplicity, we compared the intrinsic carrier concentrations of different (n, m) SWCNTs, which can be calculated by integrating density of state in conduction band (for electron) and the Fermi Dirac distribution. For the typical

中国科学院物理研究所

Institute of Physics CAS

density of states (DOS) of an SWCNT such as (7, 3), which is shown in **Figure 4c**, the intrinsic carrier concentration dominated by the first conduction band area can be described by⁴⁸

Issues that have been fixed:

Comments 1-3: Surfactant exchange. The new data showing spectral shift after replacement of DOC with SDS are a nice addition to the paper and convincingly show that surfactant exchange has happened.

Reply 1-3: Thank you for your positive comments.

Comments 1-4: Percolation. The length and percolation measurements are good additions. Convincing that percolation behaviors not responsible for observed trends.

Reply1-4: Thank you for your positive comments.

Comments 1-5: Diameter. The new diameter data are good additions. Convincingly show there is no major difference.

Reply 1-5: Thank you for your positive comments.

REVIEWER COMMENTS

Reviewer #1 (Remarks to the Author):

There is still a major error regarding use of the intrinsic carrier concentration. I have tried to point this out error in each of my previous reviews. It is discouraging to see the reviewers discounting this point. There must be some miscommunication?

From the response letter:

As stated in the previous response letter, under an applied gate voltage in a classical MOS structure, the carrier concentration (N) in the on-state can be described as [Liu Enke, et al., The physics of semiconductors (7th Edition), Publishing house of electronics industry, 2013]:

$$N \propto N_i \cdot e^{\left(\frac{|\phi_s|}{2kT} \right)}$$

Where N_i is the intrinsic carrier concentration, k is Boltzmann constant, T is temperature, and ϕ_s is surface potential in semiconductor which is determined by applied gate voltage and gate capacitance. In our present work, due to the same transistor structure and applied gate voltage, the surface potential ϕ_s are identical for different (n, m) SWCNTs. Based on equation 1, the difference in the carrier concentration of distinct (n, m) SWCNTs in the on-state is determined by that of intrinsic carrier concentration. Therefore, in main manuscript, we employ the intrinsic carrier concentration of different (n, m) SWCNTs to explain the differences in their electrical properties. In order to make it easier to understand, the corresponding discussion was added in the revised manuscript.

The problem with this response is the statement that “the surface potential ϕ_s are identical for different (n, m) SWCNTs”. This is not true – leading to major errors in interpretation in the paper. In the sub-threshold regime, the surface potential will indeed equal the gate potential for all nanotubes and thus the statement that “the surface potential ϕ_s are identical for different (n, m) SWCNTs” will be correct. But the paper is not analyzing the sub-threshold regime! Rather, in the on-regime (the regime being analyzed in the paper), the surface potential will not equal the gate potential. Instead, the surface potential will be approximately pinned near the valence or conduction band edges. Thus, the surface potential for each nanotube will be different because the bandgap of each nanotube will be different. In actuality, it will turn out to be that the carrier concentration in the on-state will be determined mostly by the 1) gate potential and 2) the gate-nanotube capacitance which varies with nanotube diameter. This is intuitive and very close to common sense – that the free carrier concentration is roughly gate capacitance times gate voltage and not linked to intrinsic carrier concentration.

See Fig. 7 "Carrier density and quantum capacitance for semiconducting carbon nanotubes" J. Appl. Phys. 104, 064515 (2008); <https://doi.org/10.1063/1.2986216> for more information. Or see attached JPG of this figure.

Response to Reviewers:

Reviewer #1:

There is still a major error regarding use of the intrinsic carrier concentration. I have tried to point this out error in each of my previous reviews. It is discouraging to see the reviewers discounting this point. There must be some miscommunication?

From the response letter:

As stated in the previous response letter, under an applied gate voltage in a classical MOS structure, the carrier concentration (N) in the on-state can be described as [Liu Enke, et al., The physics of semiconductors (7th Edition), Publishing house of electronics industry, 2013]:

$$N = N_i * e^{|\varphi_s|/2kT}$$

Where N_i is the intrinsic carrier concentration, k is Boltzmann constant, T is temperature, and φ_s is surface potential in semiconductor which is determined by applied gate voltage and gate capacitance. In our present work, due to the same transistor structure and applied gate voltage, the surface potential φ_s are identical for different (n, m) SWCNTs. Based on equation 1, the difference in the carrier concentration of distinct (n, m) SWCNTs in the on-state is determined by that of intrinsic carrier concentration. Therefore, in main manuscript, we employ the intrinsic carrier concentration of different (n, m) SWCNTs to explain the differences in their electrical properties. In order to make it easier to understand, the corresponding discussion was added in the revised manuscript.

The problem with this response is the statement that “the surface potential φ_s are identical for different (n, m) SWCNTs”. This is not true – leading to major errors in interpretation in the paper. In the sub-threshold regime, the surface potential will indeed equal the gate potential for all nanotubes and thus the statement that “the surface potential φ_s are identical for different (n, m) SWCNTs” will be correct. But the paper is not analyzing the sub-threshold regime! Rather, in the on-regime (the regime being analyzed in the paper), the surface potential will not equal the gate potential.

中国科学院物理研究所

Institute of Physics CAS

Instead, the surface potential will be approximately pinned near the valence or conduction band edges. Thus, the surface potential for each nanotube will be different because the bandgap of each nanotube will be different. In actuality, it will turn out to be that the carrier concentration in the on-state will be determined mostly by the 1) gate potential and 2) the gate-nanotube capacitance which varies with nanotube diameter. This is intuitive and very close to common sense – that the free carrier concentration is roughly gate capacitance times gate voltage and not linked to intrinsic carrier concentration.

See Fig. 7 “Carrier density and quantum capacitance for semiconducting carbon nanotubes” J. Appl. Phys. 104, 064515 (2008); <https://doi.org/10.1063/1.2986216> for more information. Or see attached JPG of this figure.

Reply: Thank you for your instructive comments. We agree that the description of the surface potential of carbon nanotubes in the last response letter is not accurate. To reveal the dependence of channel resistance of SWCNT transistors on their chiral structures, we calculate the relative junction conductance. The channel resistances are contributed by nanotube segments and junctions. At room temperature, SWCNTs exhibit ballistic transport over distances to a micrometer. In the present work, the average lengths of various (n, m) species (~ 230 nm) are much shorter than mean free paths. We thus assume that the resistance of the contributing SWCNT segments can be neglected, and that the resistances of junctions are dominated in the channel resistances. The SWCNT networks in the channel of transistors can be modeled to be a two-dimensional random resistor network. Junction conductance can thus be used to evaluate the difference in the channel resistances of distinct (n, m) SWCNT transistors. For this, the relative junction conductance has been calculated based on DOS and surface potentials of SWCNTs, and presented in Figure 4i. The results show that the effect of Type and chiral angles of SWCNTs on junction conductance is opposite to their effects on the measured channel resistance. With increasing diameter, the relative junction conductance increases. In one family, with increasing chiral angle, the relative junction conductance increases for Type I SWCNTs, while it decreases for

Type II SWCNTs. These results indicate that the resistance of SWCNT networks mainly derived from junction resistance, and reasonably explains the effect mechanism of the type and chiral angle of SWCNTs on the measured channel resistance.

We added the corresponding discussion and Figures into the revised manuscript and Supplementary Information.

Previous: Page 13, Lines 2-28 in the main manuscript

The channel resistance contains the intrinsic resistance and intertube contact resistance. The intrinsic resistance of SWCNTs is closely related to the carrier concentration and phonon scattering. The on-state carrier concentration (N) under an applied gate voltage can be described as ⁴⁹:

$$N \propto N_i * e^{|\varphi_s|/2kT} \quad (3)$$

Where N_i is the intrinsic carrier concentration, k is Boltzmann constant, T is temperature, and φ_s is surface potential in semiconductors which is determined by applied gate voltage and gate capacitance. Due to the same transistor structure and applied gate voltage, the φ_s is identical for distinct (n, m) SWCNTs. Therefore, the difference of intrinsic resistance is mainly determined by the intrinsic carrier concentration. For simplicity, we compared the intrinsic carrier concentrations of different (n, m) SWCNTs, which can be calculated by integrating density of state in conductance band (for electron) and the Fermi Dirac distribution. For the typical density of states (DOS) of an SWCNT such as $(7, 3)$, which is shown in **Figure 4c**, the intrinsic carrier concentration dominated by the first conduction band area can be described by ⁴⁸

$$N_i = \int_{E_{c1}}^{E_{c1}} DOS(E)f(E)dE \quad (4)$$

where $f(E)$ is the Fermi-Dirac distribution, that is, the occupancy probability of the on-state electron of energy E following the equation:

$$f(E) = \frac{1}{e^{[(E-E_f)/kT]} + 1} \quad (5)$$

in which E_f is the Fermi level of SWCNTs, T is the temperature and k is the

Boltzmann constant.

In equation 5, the integration interval is very sharp because of the van Hove singularity (vHS) characteristics of SWCNTs; therefore, it can be simplified to

$$N_i = \int_{E_{c1}}^{E_{c1}} DOS(E) dE \cdot \frac{1}{e^{[(E_{c1}-E_f)/kT]_{+1}}} \propto A * e^{\frac{-S_{11}}{kT}} \quad (6)$$

which is denoted as the relative carrier concentration, where A is the DOS area of the first conduction band and can be calculated by using the DOS distribution from a previous report (as shown in **Figure S14**).^{47,48}

Revised: Page13, Line 1-Page 15, Line 5 in the revised manuscript

The SWCNT networks in the channels of TFTs can be considered as two-dimensional random networks of conducting hollow cylinders^{45, 48, 49}. As mentioned above, the channel resistances are contributed by nanotube segments and junctions. At room temperature, SWCNTs exhibit ballistic transport over distances to a micrometer⁴⁹⁻⁵¹. In the present work, the average lengths of various (n, m) species (~230 nm) are much shorter than mean free paths⁴⁹⁻⁵¹. We thus assume that the resistance of the contributing SWCNT segments can be neglected, and that the resistances of junctions are dominated in the channel resistances, which is demonstrated by experimental measurements⁵²⁻⁵⁴. With this assumption, the SWCNT networks are modeled to be a two-dimensional random junction resistor network, in which the charge transfer across junctions is considered as a hopping process^{48, 55}. Similar to the transport of carriers within the nanotubes, their transport across the junctions may be affected by intrinsic density of states (DOS) and surface potentials (φ_s)^{45, 48, 49}. Thus, a conductance G_{ij}^{NT} across a junction ij connecting SWCNTs i and j can be described as:^{48, 56}

$$G_{ij}^{NT}(E_F) = \iint_{E_{cb}-e\varphi_s}^{\infty} G_{ij}(E_F, E_i, E_j) \cdot D_i(E_i) \cdot F(E_i, E_F) \cdot D_j(E_j) \cdot F(E_j, E_F) dE_i dE_j \quad (3)$$

$$G_{ij}(E_F, E_i, E_j) = \frac{e^2 \cdot \omega_{ij}^{symm}}{4k_B T \cdot \cosh\left(\frac{E_i - E_F}{2k_B T}\right) \cdot \cosh\left(\frac{E_j - E_F}{2k_B T}\right)} \quad (4a)$$

$$\approx \frac{e^2 \omega_{ij}^{symm}}{k_B T} \exp\left(\frac{E_F}{k_B T} - \frac{E_i + E_j}{2k_B T}\right) \quad (4b)$$

Where E_F is the Fermi energy, E_{cb} the energy of the bottom of conduction band, e the elemental charge, φ_s the surface potential created by gate voltage, E_i and E_j the energies in SWCNTs i and j , respectively, $D_i(E)$ and $D_j(E)$ are their DOS, $F(E, E_F)$ the Fermi–Dirac distribution. $G_{ij}(E_F, E_i, E_j)$ the bond conductance of junction ij at the energies of E_i and E_j for SWCNTs i and j , k_B the Boltzmann constant, T the absolute temperature, ω_{ij}^{symm} a symmetrized hopping rate between SWCNTs i and j . In equation (4),

$$\omega_{ij}^{symm} = \omega_0 \cdot \exp(-|\Delta E_{ij}|/2k_B T) \quad (5)$$

Where ω_0 a hopping prefactor determined by the charge-transfer integral between the states and ΔE_{ij} the energy difference ($E_j - E_i$) between the states j and i . Due to the use of dense networks of single-chirality SWCNTs, many nanotube junctions are present and uniformly distributed in the channels from source to drain electrodes. We assume that ΔE_{ij} is the same for each junction. And thus, the term of $e^2 \omega_{ij}^{symm}/k_B T$ in equation 2b can be considered constant. The surface potential φ_s increase with gate voltage, and approximately pinned near the conduction (or valance) band edges ($\sim E_g/2e$) in the on-regime of TFTs⁵⁷. The theoretical simulation results show that the first conduction subband determines transport performance in practical devices^{57, 58}. Actually, only states with energies close to the band edge contribute to the transport due to the rapidly decaying of Fermi tail⁵⁸. For simplicity, we only consider the first subband of the conduction band. It has been demonstrated that the energy difference between the bottoms of the first and second conduction subbands is close to $E_g/2$ ⁵⁸. With these assumptions, a conductance G_{ij}^{NT} of a typical junction ij in the on state can be calculated by substituting equation (3) with equation (4b) and

integrating from $E_{cb} - e\phi_s$ to $E_{cb} + E_g/2 - e\phi_s$ in the first conduction band. Because the value of ω_0 is unknown⁴⁸, the G_{ij}^{NT} is further simplified to a relative junction conductance $G_{ij}^{NT'}$ as follows:

$$G_{ij}^{NT} \propto G_{ij}^{NT'} = A^2 \cdot e^{(-3S_{11})/2k_B T} \quad (6)$$

Where A is the DOS area of the first conduction band and can be calculated by using the DOS distribution from a previous report (as shown in **Figure S14**)^{47,48}. A typical DOS of an SWCNT such as (7, 3) is shown in **Figure 4c**.

With equation (6), a relative junction conductance ($G_{ij}^{NT'}$) for different (n, m) can be evaluated, and accordingly, the corresponding channel resistances could be compared. As shown in **Figure 4i**, with increasing diameter, the relative junction conductance generally increases. In one family, with increasing chiral angle, the relative junction conductance increases for Type I SWCNTs, while it decreases for Type II SWCNTs. Clearly, the effects of the type and chiral angle of SWCNTs on junction resistance are opposite to their effects on the measured channel resistance, indicating that the resistance of SWCNT networks mainly derived from junction resistance.

Previous: Page 11, Figure 4i and Figure Caption in the main manuscript

(i) Plots of the relative carrier concentration as a function of SWCNT diameter deduced from equation 5 at room temperature ($kT = 26$ meV).

Revised: Page 11, Figure 4i and Figure caption in the revised manuscript

(i) Plots of the relative junction conductance G_{ij}^{NT} as a function of SWCNT diameter deduced from equation (6) at room temperature ($k_B T = 26$ meV).

Previous: Supplementary Information, Figure S15

The relationship between the channel resistance and carrier concentration.

Figure S15. The relationship between the channel resistance and the carrier concentration $A \cdot \exp(-\text{gap}/kT)$. The results show that the channel resistance of the same-family SWCNTs decreases with increasing the carrier concentration. Error bars in Figure are the standard deviation of statistics.

Revised: Supplementary Information, Figure S15:

The relationship between the channel resistance and the relative conductance.

Figure S15. The relationship between the channel resistance and the relative junction conductance $A^2 \cdot \exp(-3S_{11}/2k_B T)$. The results show that the channel resistance of the same-family SWCNTs decreases with increasing the relative junction conductance. Error bars in Figure are the standard deviation of statistics.

REVIEWERS' COMMENTS

Reviewer #1 (Remarks to the Author):

The previous major mistake in analysis using semiconductor theory is gone. It has been replaced by a much more convoluted analysis of nanotube-nanotube junction resistance. I cannot tell if the new analysis has a strong foundation in reality or not because it is too convoluted and somehow a $3/2$ S_{11} dependence is produced. But, at least it does not have any glaring mistakes.

As stated in the previous reviews, the experiment data are interesting and should be published. No very strong theory / analysis of the data is ever firmly established to explain the data but the paper should still be published so others can start to think about what the data might mean.

中国科学院物理研究所

Institute of Physics CAS

Response to Reviewers:

Reviewer #1:

The previous major mistake in analysis using semiconductor theory is gone. It has been replaced by a much more convoluted analysis of nanotube-nanotube junction resistance. I cannot tell if the new analysis has a strong foundation in reality or not because it is too convoluted and somehow a $3/2$ S_{11} dependence is produced. But, at least it does not have any glaring mistakes.

As stated in the previous reviews, the experiment data are interesting and should be published. No very strong theory/analysis of the data is ever firmly established to explain the data but the paper should still be published so others can start to think about what the data might mean.

Reply: Thank you for your positive comments and valuable recommendations. The coefficient $3/2$ is the result of integrating equation (3).